# Visualizing the membrane disruption action of antimicrobial peptides by cryo-electron tomography

Eric H.-L. Chen[1,10], Chun-Hsiung Wang [1,10], Yi-Ting Liao [1,10], Feng-Yueh Chan [2], Yui Kanaoka[2], Takayuki Uchihashi [2,3,4], Koichi Kato[4,5,6], Longsheng Lai[7], Yi-Wei Chang [7], Meng-Chiao Ho [1,8,9] ✉ & Rita P.-Y. Chen [1,8] ✉

The abuse of antibiotics has led to the emergence of multidrug-resistant microbial pathogens, presenting a pressing challenge in global healthcare. Membrane-disrupting antimicrobial peptides (AMPs) combat so-called superbugs via mechanisms different than conventional antibiotics and have good application prospects in medicine, agriculture, and the food industry. However, the mechanism-of-action of AMPs has not been fully characterized at the cellular level due to a lack of high-resolution imaging technologies that can capture cellular-membrane disruption events in the hydrated state. Previously, we reported PepD2M, a de novo-designed AMP with potent and wide-spectrum bactericidal and fungicidal activity. In this study, we use cryo-electron tomography (cryo-ET) and high-speed atomic force microscopy (HS-AFM) to directly visualize the pepD2M-induced disruption of the outer and inner membranes of the Gram-negative bacterium *Escherichia coli*, and compared with a well-known pore-forming peptide, melittin. Our high-resolution cryo-ET images reveal how pepD2M disrupts the *E. coli* membrane using a carpet/detergent-like mechanism. Our studies reveal the direct membrane-disrupting consequence of AMPs on the bacterial membrane by cryo-ET, and this information provides critical insights into the mechanisms of this class of antimicrobial agents.

The COVID-19 pandemic has underscored the societal risks of infectious diseases for which ineffective treatments exist[1]. A similar threat is posed by the emergence of multidrug-resistant microbes[2], and, moreover, increased antimicrobial resistance has been detected during the COVID-19 pandemic due to the overuse of antibiotics[2–4]. Thus, new strategies and approaches that can help to overcome the challenges presented by antimicrobial resistance are needed[5–7]. Natural antimicrobial peptides (AMPs) are a self-defense mechanism of organisms and have potent inhibitory effects against a broad spectrum of microorganisms, including bacteria, fungi, and viruses. To date,

[1]Institute of Biological Chemistry, Academia Sinica, Taipei 11529, Taiwan. [2]Department of Physics, Nagoya University, Nagoya 464-8602, Japan. [3]Institute for Glyco-core Research (iGCORE), Nagoya University, Nagoya 464-8602, Japan. [4]Exploratory Research Center on Life and Living Systems (ExCELLS), National Institutes of Natural Sciences, Okazaki, Aichi 444-8787, Japan. [5]Institute for Molecular Science, National Institutes of Natural Sciences, Okazaki, Aichi 444-8787, Japan. [6]Graduate School of Pharmaceutical Sciences, Nagoya City University, Nagoya 467-8603, Japan. [7]Department of Biochemistry and Biophysics, Perelman School of Medicine, University of Pennsylvania, Philadelphia, PA 19104-6059, USA. [8]Institute of Biochemical Sciences, National Taiwan University, Taipei 10617, Taiwan. [9]Institute of Biochemistry and Molecular Biology, College of Medicine, National Taiwan University, Taipei 100, Taiwan. [10]These authors contributed equally: Eric H.-L. Chen, Chun-Hsiung Wang, Yi-Ting Liao. ✉e-mail: joeho@gate.sinica.edu.tw; pyc@gate.sinica.edu.tw

over 3300 AMPs have been reported in the antimicrobial peptide database (APD3). The functions of AMPs include disrupting microbial cell membranes, modulating the immune response of the host, and regulating inflammation[8]. In this regard, AMPs are promising therapeutic candidates because their antimicrobial mechanisms are different from conventional small-molecule drugs, which typically target cellular processes (e.g., protein or cell-wall synthesis)[9]. Further, a critical advantage of AMPs is that many of them directly disrupt the membrane, and their action time is short (within hours)[10–12]; hence, it is not easy for microbes to develop resistance to them[13].

To explain the mode of action of AMPs, various models have been proposed based on spectrometric data and molecular-dynamics simulations using artificial membrane bilayers or liposomes[14–21]. In general, the models can be classified as either pore-forming or non-pore-forming. The two pore-forming models, in which transmembrane pores form on the lipid bilayer, are the barrel-stave model and the toroidal model[16,22–24]. The difference between these two models is that AMPs interact with the fatty acyl chains surrounding the pore wall in the former, whereas AMPs only interact with the lipid head groups in the latter. One famous AMP classified in the pore-forming model is melittin, which is a 26-residue natural peptide present in bee venom. On the other hand, many non-pore-forming models have also been proposed; prominent among these are the carpet model and the detergent-like model[18,25,26]. In the carpet model, AMPs interact only with the lipid head group and lie on the membrane surface. In the detergent-like model, lipids are dissolved and taken away from the lipid bilayer by AMPs in a detergent-like manner. Depending on peptide concentration, conditions, and lipid composition, one AMP may follow more than one mechanism in disrupting membranes[22]. The working models of most AMPs have been studied using an artificial membrane, while there has been a lack of research on AMP actions on genuine membranes.

Recent advances in cryo-electron tomography (cryo-ET), and the corresponding image analyses, provide an excellent method for studying the three-dimensional (3D) structure of living cells[27–30]. Distinct from scanning electron microscopy (SEM), which can only probe the surface of a cell, and transmission electron microscopy (TEM), which provides a two-dimensional (2D) projection of a thin cell slice, 3D tomographic reconstruction through cryo-ET provides a whole-volume image of cells. Cryo-ET data can be visualized either as 2D tomographic slices or as 3D surfaces, allowing the observation of detailed morphological changes in membranes. Most importantly, the cryo-ET samples are preserved in their frozen-hydrated (native-like) state, providing an opportunity to capture membrane disruption at the moment when the sample is frozen. While cryo-ET offers an attractive solution for understanding cellular-membrane structures in the frozen-hydrated state, an important technical challenge must be overcome: the thickness of most bacteria exceeds the ~500 nm practical limit of cryo-ET using conventional electron microscopes[31,32]. This limit can be circumvented by using cryo-focused ion beam (cryoFIB) milling to thin down the sample[32–35]. Unfortunately, it is challenging to use cryoFIB to section out the exact membrane region of cells that has been damaged by the AMP treatment for cryo-ET imaging.

One alternative strategy is to use bacterial minicells, which can be produced from E. coli deficient in MinCDE genes[36]. The proteins encoded by these genes are responsible for positioning the Z-ring at the center of the cell during binary cell division. Without these proteins, the cells cannot locate the cell division plane; thus, a small portion is pinched off from the cell poles, forming minicells. E. coli minicells do not have chromosomal DNA but have the same cytosolic materials and membrane structure as normal E. coli cells[37,38]. Minicells that have a diameter of <500 nm can be separated from normal E. coli cells by centrifugation or filtration, offering a valuable solution for overcoming the thickness limitations imposed by cryo-ET. The combination of cryo-ET and minicells has been previously used for studying the molecular architecture of chemosensory systems, protein secretion, and DNA translocation[39–44].

Atomic force microscopy (AFM) has a lateral resolution of 1–5 nm and a height resolution of 1 Å and is suitable for monitoring the action modes of AMPs for membranes[45–52]. The development of high-speed atomic force microscopy (HS-AFM) enhanced the temporal resolution of this technique and enabled the direct visualization of the dynamic motions of AMPs at high spatiotemporal resolution[21,53–57].

In this work, we describe the use of cryo-ET and HS-AFM to study how a de novo-designed peptide, pepD2M, interacts with the natural membrane of Gram-negative bacteria and compare it with the action of melittin. PepD2M is a 14-residue amphipathic peptide that can destroy the cell membrane of bacteria and fungi (including spores and mycelia)[58]. Its bactericidal activity is similar to melittin, while its hemolytic activity is 128 times lower than melittin[11,12]. We find that pepD2M severely disrupts the E.coli membrane, and the removed lipids form many lipid clusters. It could lead to the formation of pores on the membrane, but the pore size was bigger than that formed by melittin. On the other hand, melittin created many small pores on the E. coli membrane and cell shrinking and, especially, can induce blister formation on the outer membrane. Our findings reveal 3D membrane morphology changes caused by different types of AMPs with a nanometer-range resolution and provide critical insights into the mode of action of these antimicrobial molecules.

## PepD2M specifically targets bacterial membrane-like liposomes

PepD2M is a de novo-designed amphipathic peptide with only 14 residues, of which 57% of them are positively charged[12,58]. As shown in the circular dichroism (CD) spectra (black line in Fig. 1a, b), pepD2M forms a random coil structure in water (indicated by a negative peak at -196 nm) but turns into an α-helical structure in phosphate buffer (indicated by the two negative peaks at 208 and 220 nm). The data suggested that salt promoted the helix formation of the peptide. We used the DOPC liposomes to mimic the neutral mammalian cell membrane and PE/PG liposome (POPE:DOPG = 1:1, w/w) to mimic the negatively charged bacterial membrane. When pepD2M was mixed with the negatively-charged PE/PG liposomes in water (red line in Fig. 1a), the CD spectrum was red-shifted, indicating that there was an interaction between the peptide and the lipids. Interestingly, when the PE/PG liposomes were prepared in phosphate buffer, the addition of pepD2M resulted in the solution immediately becoming turbid, and the CD ellipticity was close to zero (red line in Fig. 1b). The low CD signal is because pepD2M co-precipitated with the lipids, as shown in the cuvette in Fig. 1b. This result suggested that salt promoted the lipid-removing effect of pepD2M on the PE/PG liposomes. In contrast, when the neutral DOPC liposomes were tested, no matter whether the liposomes were prepared in water or buffer, the resulting CD spectra (green line in Fig. 1a, b) were the same as those without liposomes (black line in Fig. 1a, b). This result suggested that pepD2M could not interact with DOPC at all. Consistent with our conclusion, in the fluorescent leakage assay, pepD2M caused dye to be released from the PE/PG liposomes, while melittin caused dye to be released from the DOPC liposomes (Supplementary Fig. 1).

## Initial characterization of E. coli minicells by cryo-ET focused on membrane structures

As shown in Fig. 2a, b, the minicells were evenly distributed on the cryo-EM grids. Due to defective cell division, some minicells possessed an enlarged periplasm, and their cytoplasm was not in the center (Fig. 2c). Tomographic reconstructions revealed the prominent features of these minicells, as shown in Fig. 2d, e, and Supplementary Fig. 2, in which the outer membrane (OM), peptidoglycan (PG), and inner membrane (IM) are resolved. The PG layer maintains a close and fixed distance (10.7 ± 0.1 nm, Fig. 2f) from the OM layer, indicating that it is covalently linked to the OM but not the IM.

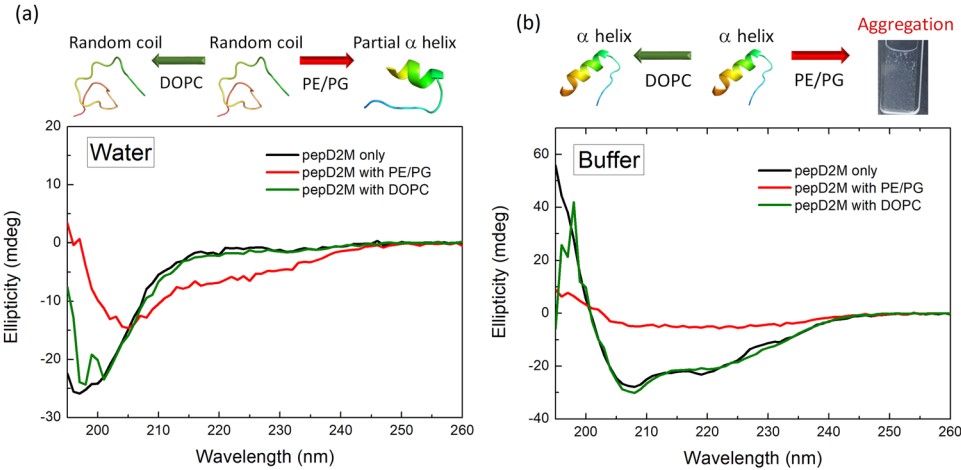

**Fig. 1 | Lipid-induced structural change of pepD2M.** CD spectra of pepD2M in water (**a**) and in buffer (**b**). Black line: without liposome; red line: mixed with the PE/PG liposomes; green line: mixed with the DOPC liposomes. Illustrations of the structural change that occurs upon conditional changes are shown above the spectra.

## Visualization of the *E. coli* minicells treated with pepD2M

We previously designed a series of amphipathic peptides with a particular sequence pattern BBHBBHHBBH (B: basic residues; H: hydrophobic residues) that showed excellent antimicrobial activity against Gram-positive and Gram-negative bacteria and fungi[12,58]. Among the designed peptides, pepD2M exhibited antimicrobial efficacy against pathogen infection *in Planta*[58]. To probe how this peptide disrupts the Gram-negative cell wall, minicells treated with pepD2M were plunge-frozen to preserve the bacterial membranes at the moment of disruption. As the detailed morphological changes were preserved at the very moment of freezing, we could successfully observe several different changes. A series of cryo-ET micrographs of the pepD2M-treated minicells were then captured by tilting the sample along its axis. Next, the 3D tomograms were reconstructed using the back-projection method[59,60]. After reconstruction, the two-dimensional (2D) slices through the whole-cell volume allowed us to visualize the membrane changes and identify the areas of interest for further inspection. The tomogram movies of one intact minicell (Supplementary Movie 1) and two minicells treated with pepD2M (Supplementary Movies 2, 3) are provided in the supplementary materials.

After treatment with pepD2M, several phenomena were observed and classified as five morphological changes: Change 1: Irregular holes formed on the OM (indicated by a yellow arrow in Fig. 2g); Change 2: Minicell shrinkage (indicated by a red arrow in Fig. 2h); Change 3: Part of the membrane had a lower density (indicated by blue arrows in Fig. 2i), which might be because the peptide removed the lipids and decreased the lipid density on the membranes; Change 4: The lipids removed from the minicell membrane collapsed into lipid clusters (indicated by black arrowheads in Fig. 2h–l); Change 5: The remaining parts of the minicells (indicated by white arrows in Fig. 2j–l) were trapped in the clumps of lipid clusters. We collected 123 tomograms, and these five changes were counted (Fig. 2m). Our results showed that pore formation and membrane with a lower density were observed at a low peptide concentration (16 μg/mL). In contrast, when higher peptide concentrations were used, shrunken cells and lipid clusters with or without residual cells trapped inside were frequently found (Fig. 2n).

Moreover, we could identify different areas of interest in different 2D slices (Fig. 3). For example, the minicell in Fig. 2g had a membrane that was disrupted but the cytosolic material from it had not been released yet. The tomographic slices of this minicell at different z-axis values are shown in Fig. 3a–d. On the membrane of this damaged

minicell, three holes could be clearly observed on the OM, while the IM remained intact (inset in Fig. 3a). From a different slice view of the same minicell, the periplasm was enlarged, while the PG layer maintained a close and fixed distance from the OM layer (Fig. 3b). A large portion of the OM had diffused away, as shown in Fig. 3c. Figure 3d shows two volcanic-crater-like holes in the OM and the release of the periplasmic materials. The reconstructed 3D tomogram of this minicell with computational segmentation is shown in Fig. 3e. Figure 3f displays another minicell treated with pepD2M, and the images clearly show that membrane disruption occurred at multiple sites in this case. A large pore with a diameter of 45 nm was seen on the left side (indicated by a purple arrow, Fig. 3f). Both the OM and IM were damaged by pepD2M, and the damaged regions are not correlated. A piece of the broken OM layer was pushed outwards (indicated by a red arrowhead in Fig. 3f).

## Dynamics of membrane disruption of pepD2M by HS-AFM

The cryo-ET images clearly indicate that pepD2M disrupts minicells by taking away membrane lipids; however, we wanted to investigate the dynamics of membrane disruption at the time scale of seconds, so we conducted HS-AFM experiments.

The experiments were conducted by adding 16 μg/mL pepD2M to lipid bilayers extracted from *E. coli* adsorbed on a mica surface. The formation and disappearance of dimples on the membrane were rapidly captured using HS-AFM (Fig. 4a, b). Movies of the pore formation acquired using HS-AFM are provided in the supplementary materials (Supplementary Movies 4, 5). Pores appeared in the lipid membrane about ten seconds after pepD2M injection (indicated by arrowheads in Fig. 4a, b), and their depth and size varied with time. Some pores could be refilled by the surrounding lipid membrane, while others could not. The formation and disappearance of these pores on the lipid membrane occurred randomly at various locations.

Figure 4c shows the time evolution of the pore depth and size for seven pores (#1–#7). Our analysis showed that the AMP-induced pore formation in the membrane is highly dynamic. In the short duration after peptide addition, Pore #4 repeatedly formed and disappeared at the same location (Supplementary Movie 4). On the other hand, the depth of Pore #2 was almost constant, but the size gradually increased. The maximum depth of Pore #1–#5 is about one nanometer. The small depth indicates that these pores did not pierce the membrane (the thickness of the lipid membrane is about four nanometers). Therefore, they are actually dimples rather than pores. On the other hand, after

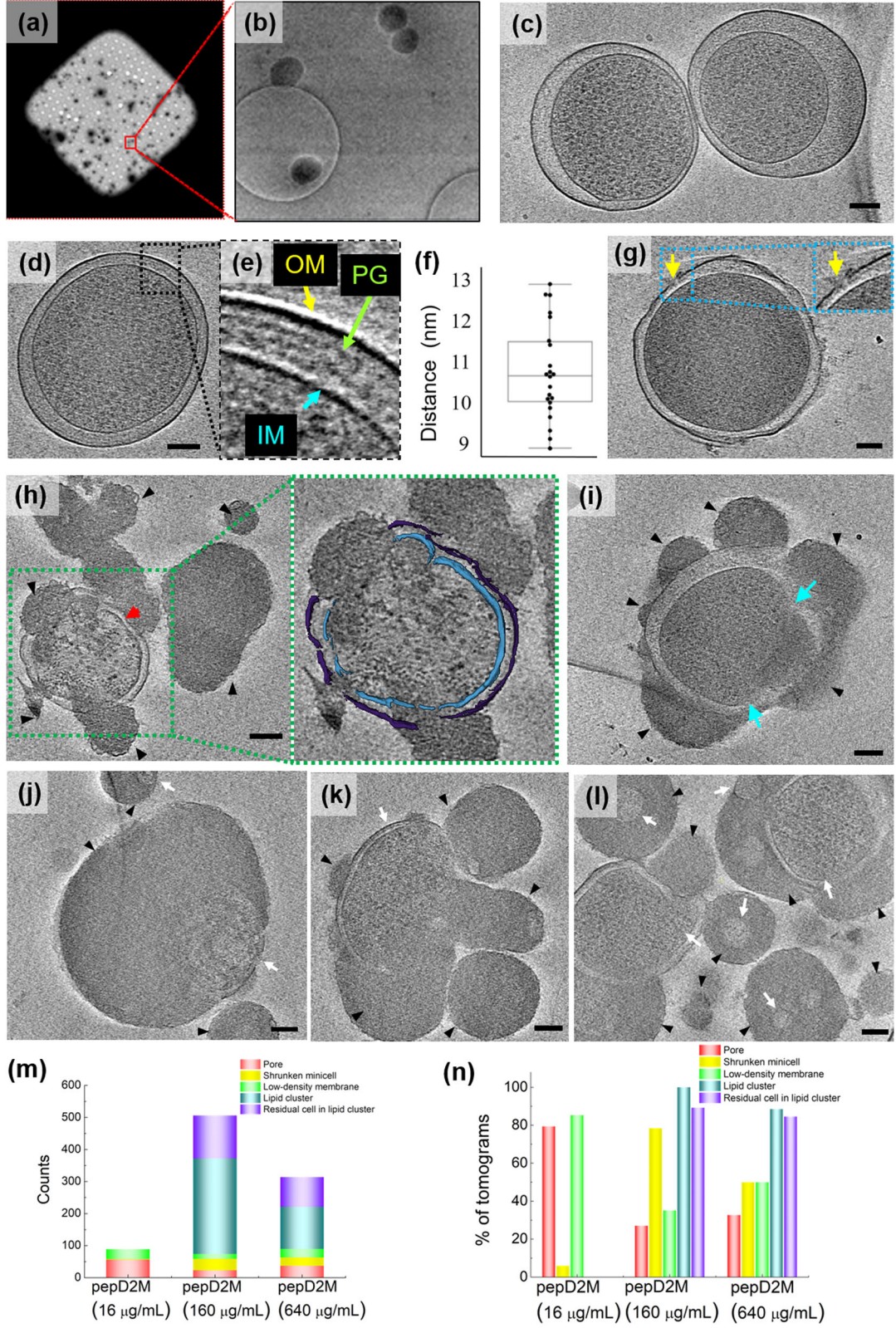

more than five minutes of injecting pepD2M into the solution, the depth and size of the pores formed in the membrane increased (Pores #6 and #7), with the maximum depth being about four nanometers, suggesting that they pierced the membrane. Even in such cases, the pore's depth and size fluctuate with time. The lipids removed by the peptide gradually deposited on the free mica surface (Fig. 4d; Supplementary Movie 6).

## Cryo-ET images of the *E. coli* minicells treated with melittin

Cryo-ET images of minicells treated with a well-known pore-forming AMP, melittin, were also obtained for comparison with the action of pepD2M[23]. A previous biophysical study showed that the interaction between melittin and lipids depends on the peptide/lipid (P/L) ratio[22]. At a low P/L ratio, melittin lies on the membrane surface parallel to the membrane. When the P/L ratio increases, melittin inserts into the

**Fig. 2 | Cryo-ET studies of minicells with and without pepD2M treatment. a** A representative square of a Quantifoil R 2/2 Holey carbon grid loaded with minicells. **b** A zoomed-in image of the red box in (**a**). Minicells could be in a hole or on the carbon film. **c** Two untreated minicells with an enlarged periplasm. **d** A tomographic slice of an untreated minicell. **e** A zoomed-in image of the blue box in (**d**). OM: outer membrane; PG: peptidoglycan; IM: inner membrane. **f** The average distance between the OM and PG is 10.7 ± 0.1 nm (mean ± standard deviation). Eleven minicells were measured. Each minicell was measured twice at different locations. The plot is presented as 25th percentile, median, and 75th percentile, with Tukey

whiskers representing ±1.5 × the interquartile range. **g–l** Tomographic slices of minicells treated with pepD2M to show examples of five morphological changes. **g** Change 1: pore formation (yellow arrow). **h** Change 2: shrunken cell (red arrow). **i** Chang 3: low-density membrane (blue arrow). **j–l** Change 4: lipid cluster (black arrowhead); Change 5: lipid cluster with cell residuals inside (white arrow). The scale bar represents 100 nm. The insets in **g** and **h** are the enlarged image from the box. IM: cyan; OM: violet. **m** Counts of changes 1–5 after the treatment with different concentrations of pepD2M. **n** The proportion of Changes 1–5 occurrence in 123 tomograms.

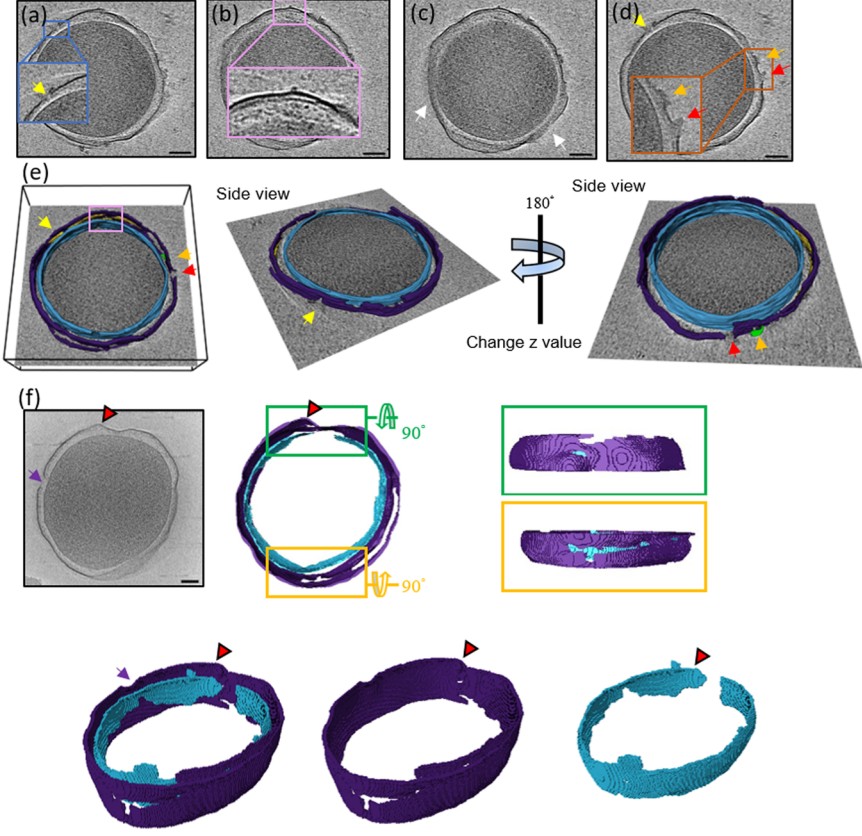

**Fig. 3 | Cryo-ET slices of two minicells treated with pepD2M and their inner and outer membranes. a–d** The cryo-tomographic slices of the minicell in Fig. 2g at different z-axis values. The insets present the zoomed-in images from the indicated boxes. Pores are pointed by arrows with different colors. Two volcanic-crater-like pores are indicated by the red and orange arrows. **e** 3D tomogram of the minicell in

(**a–d**). **f** A cryo-tomographic slice (left) and its corresponding segmentations (right) of another minicell after the peptide treatment. A large pore with a diameter of 45 nm is indicated by a purple arrow. IM: cyan; OM: violet; PG: yellow; released material: green. Scale bar = 100 nm.

membrane perpendicularly and forms a pore. The cryo-ET images of the melittin-treated minicells did show many pores with the released cytosolic materials nearby (indicated by red arrowheads in Fig. 5a, b; Supplementary Movie 7). Interestingly, several minicells showed a raspberry-like morphology, with many blisters (indicated by white arrowheads in Fig. 5c) protruding from the OM, while the IM remained unchanged (See Supplementary Movie 8). The blister formation was probably due to the membrane-thinning effect caused by melittin lying on the membrane[23], although we did not find that the number of the minicells with blisters is correlated to the peptide concentration (Fig. 5g, h). At higher peptide concentrations, a large number of shrunken minicells that still contained double membranes were observed (indicated by yellow arrowheads in Fig. 5d–f). These shrunken minicells adhered together and formed clusters of residual minicells (Fig. 5f). Unlike pepD2M, there are very few lipid clusters in the melittin treatment. Only several small lipid clusters (indicated by black arrowheads in Fig. 5b) could be observed. Figure 5g shows the counts of pores, minicells with blisters, shrunken minicells, and lipid clusters

at different peptide concentrations. Their occurrence percentage in 140 tomograms is displayed in Fig. 5h.

## Dynamics of membrane disruption of melittin by HS-AFM

For comparison, we also recorded HS-AFM images of the *E. coli* lipid bilayers treated with melittin. Using the same peptide concentration, melittin formed pores slower (after 553 s) than pepD2M (after 16 s; Fig. 6a; Supplementary Movie 9). The pore depth and size of five pores (#8–12) are shown in Fig. 6b. We could also see that a pore appeared and disappeared (Pore #10), and the lipids deposited on the free mica surface (Fig. 6c). The maximum pore depth for most of the pores was around 1 nm, suggesting that they might be dimples caused by the membrane-thinning effect of melittin rather than toroidal pores which transverse across the membrane. We did not observe deep lipid penetration (>3 nm) and large pores (>200 nm²), as seen in Pore #6 and #7 in the pepD2M treatment (Fig. 4c).

One could see that melittin cannot take away lipids as efficiently as pepD2M. Using Triton X-100 as a positive control, we found that this

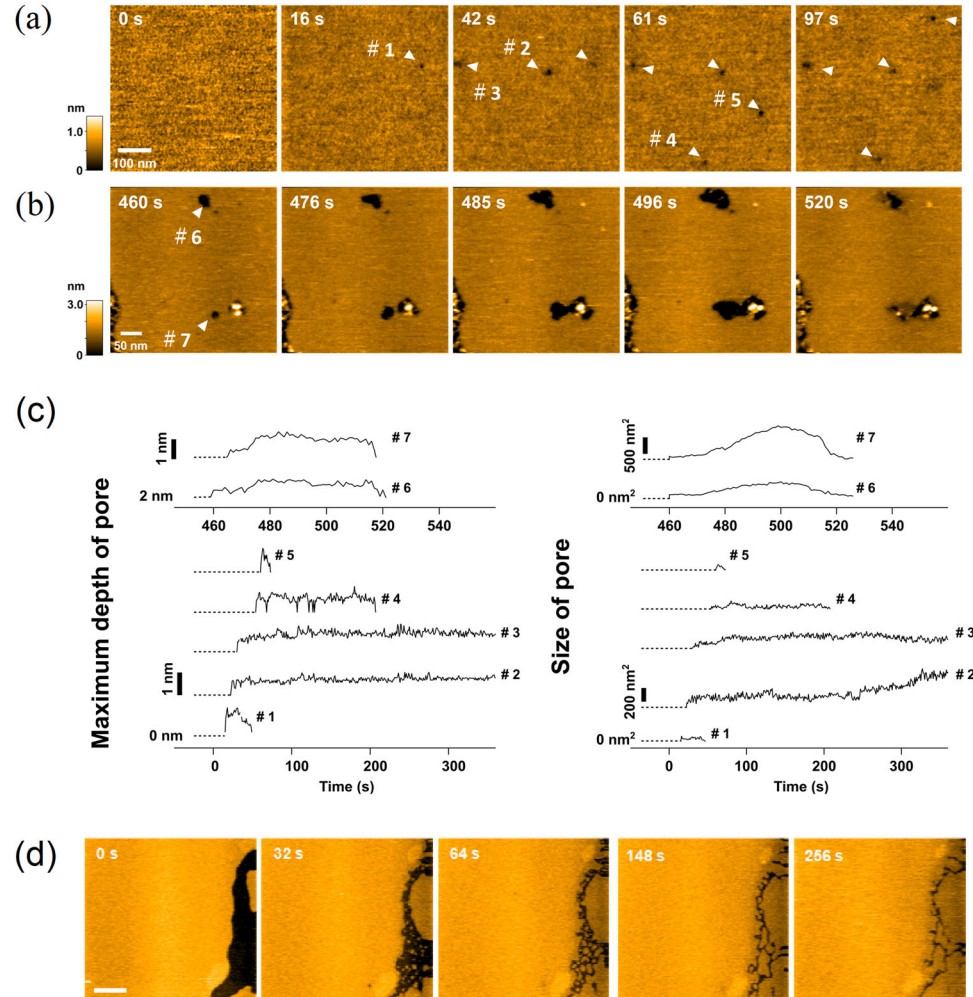

**Fig. 4 | Lipid-removing effect of pepD2M observed by HS-AFM. a, b** HS-AFM images of extracted *E. coli* lipids immediately after pepD2M treatment (16 μg/mL). The scanned regions in **a** and **b** are different. The detection time is indicated on the image. **c** Time evolution of the pore depth and size of the seven pores shown in (**a**) and (**b**). **d** HS-AFM images of the lipid deposition on the free mica surface after pepD2M treatment. See Supplementary Movies 4–6.

detergent removed a significant amount of lipid molecules from the membranes of minicells, leaving a discontinuous OM (Supplementary Fig. 3; Supplementary Movie 10). However, the lipids were well-dissolved in the detergent so that no lipid clusters could be found.

## Discussion

The goal of this work was to directly observe the membrane disruption action of AMPs on the Gram-negative bacterial membrane. This was achieved by treating *E. coli* minicells with AMPs, preserving them in the vitrified state, capturing images of them using cryo-ET, and conducting tomographic reconstruction for a detailed inspection of morphological changes that occurred.

PepD2M is an amphipathic peptide. When it forms an α-helical structure, the positively charged Lys covers ~180 degrees of the helical surface. Although this design seems to favor transmembrane pore formation (the hydrophobic part interacts with the fatty acyl chains, while the hydrophilic part forms the pore wall), our experimental data show that this is not the case and indicate the process as follows. First, pepD2M removes the lipids in the OM, leading to the formation of irregular holes (Figs. 2g, 3a) and periplasmic material release (Fig. 3d). A significant portion of the IM is removed, suggesting that the action of pepD2M does not need the negative charge of the lipopolysaccharide that is present only in the OM (Fig. 3f). Once holes are created in the IM, the cytoplasmic materials within the cells are released, and the cells shrink (Fig. 2h). Finally, pepD2M and the membrane lipids collapse into

lipid clusters (Fig. 2h–l). Due to hydrophobic and electrostatic interactions, the lipid clusters can adhere to the cell surface, which might prevent further leakage of cellular content if the adhere site has a pore (Fig. 2h, l). These lipid clusters were found to be filled with electron-dense materials. Figure 1b shows that pepD2M precipitated together with the negatively-charged lipid. These lipid clusters, observed by cryo-ET, might comprise aggregated peptides, lipids, and released cytosolic materials. Compared to melittin, pepD2M produced a significant amount of lipid clusters, with many small residual minicells trapped inside (Fig. 2l–n). In contrast, melittin produced a considerable number of shrunken minicells that were still protected by a double membrane (Fig. 5f–h). The formation of these lipid clusters is consistent with the rough surface of AMP-damaged bacteria and the surrounding debris observed by AFM[46,47,54] and SEM[11,61]. The crater-like membrane with the volcanic release of periplasmic materials is similar to the TEM image of detergent-like human apolipoprotein L-treated *E. coli*[62]. Although similar changes have been reported using SEM or TEM, crucially, cryo-ET provides higher resolution imaging and whole-cell-volume 3D images to visualize morphological changes, more importantly, in a native-like state. The cryo-ET images clearly demonstrate that membrane disruption caused by pepD2M is distinct from that achieved using melittin.

When comparing the pore size observed through cryo-ET (Fig. 7a), the diameters of the pores created by pepD2M had considerable variation (all the sites with missing membranes were counted

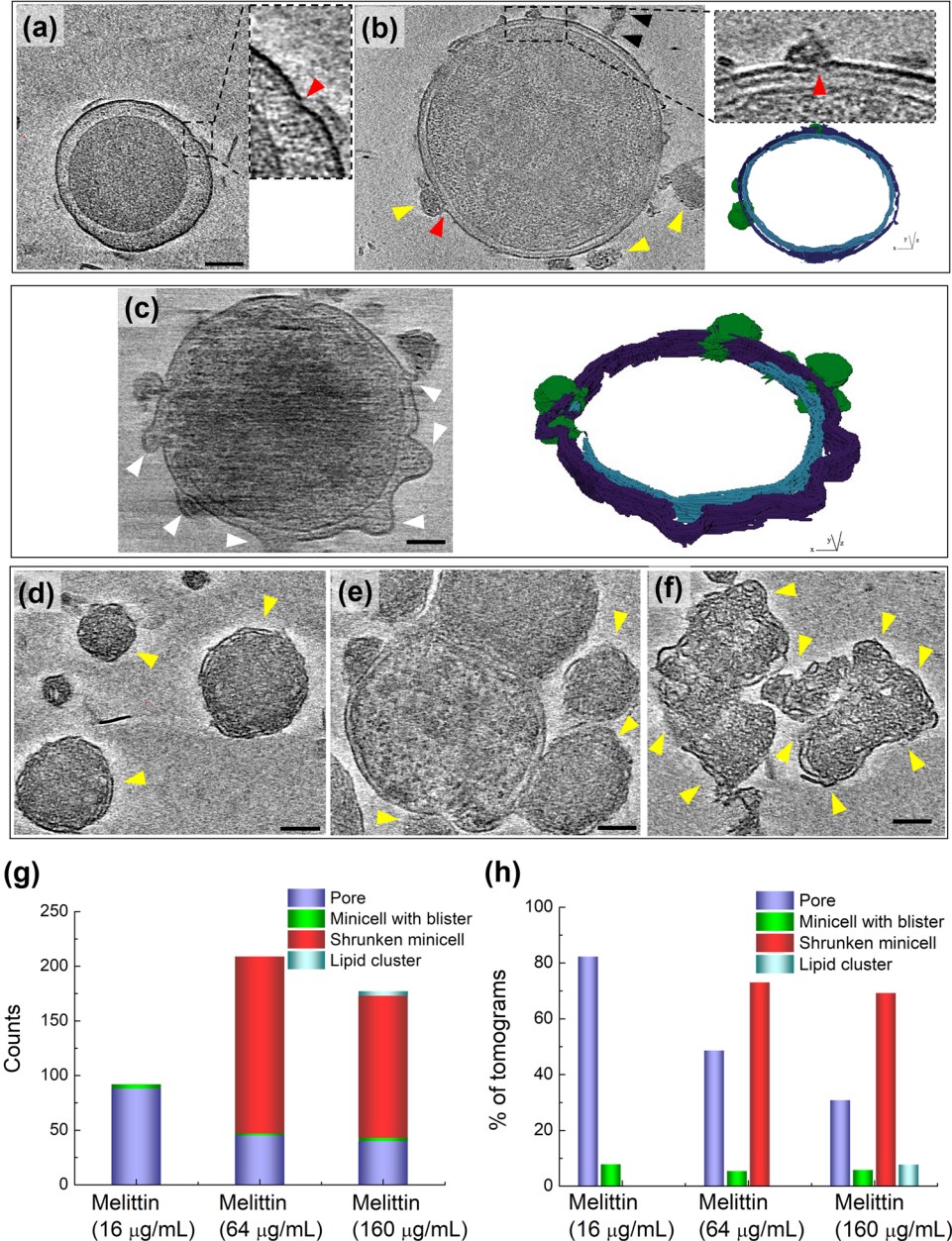

**Fig. 5 | Cryo-ET results of minicells treated with melittin. a–f** The cryo-tomographic slices and the 3D segmentation of the minicells. IM: cyan; OM: violet; released material: green. Insets in (**a**) and (**b**): enlarged images of the membrane with a pore. The red arrowhead indicates the location of the pore. Lipid clusters are indicated by black arrowheads. The location of blisters and shrunken minicells are indicated by white and yellow arrowheads, respectively. Scale bar = 100 nm. **g** Counts of pores, minicells with blisters, shrunken minicells, and lipid clusters after the treatment with different concentrations of melittin. **h** The proportion of the occurrence of pores, minicells with blisters, shrunken minicells, and lipid clusters in 140 tomograms.

as pores) at high concentrations (160 and 640 µg/mL). At the same weight concentration, the pore sizes created by pepD2M were more extensive than those produced by melittin, suggesting the strong membrane disruption effect of pepD2M. When comparing the pore area observed by HS-AFM (Fig. 7b), melittin did not show concentration-dependent variation. On the other hand, low-concentration pepD2M could not form any pores, while high-concentration pepD2M could create big pores (200–1000 nm²) in the lipid bilayers. We did not compare the pore size observed using cryo-ET and HS-AFM because cryo-ET watched the pore formed on minicells where osmotic pressure plays a role in the membrane damage. According to the pore depth from the HS-AFM data, both peptides created shallow dimples on lipids, but high-concentration pepD2M had the chance to remove the lipid bilayer (Fig. 7c).

PepD2M contains a particular sequence pattern BBHBBHHBBH (B: basic; H: hydrophobic)[58]. When the peptide forms an α-helical structure, the basic residues are arranged as a triangle-shaped charge cluster to bind negatively charged lipids, like a claw in the claw machine[58]. In this study, our CD experiments show that pepD2M only interacts with the negatively charged liposomes, and an α-helical structure is induced by the peptide–lipid interaction, as described in the carpet model. The HS-AFM images clearly show the rapid lipid-removing effect of pepD2M (Fig. 4). The lipid preference suggests that this peptide design could have a better selectivity between bacterial and mammalian cell membranes (Fig. 1 and Supplementary Fig. 1).

Most importantly, this study demonstrates the peptide-induced morphological changes of bacterial membranes in a native-like state

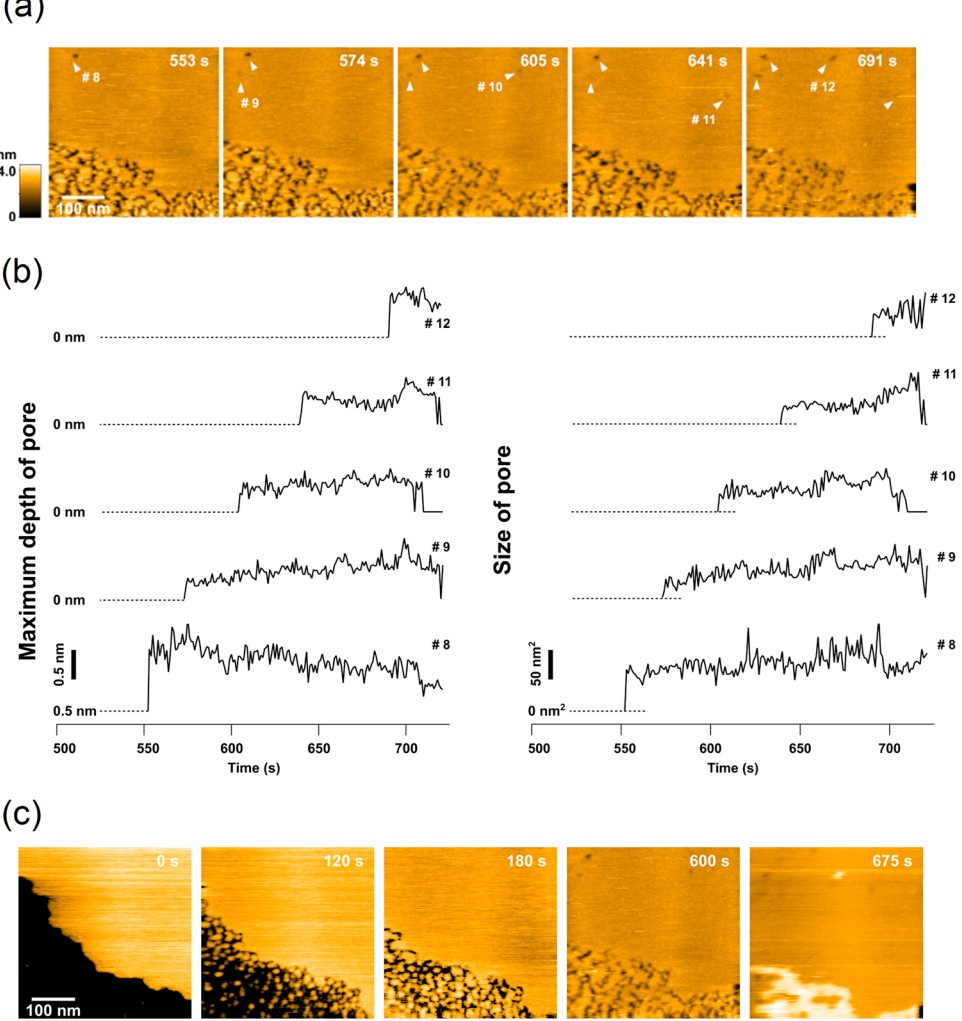

**Fig. 6 | Lipid-removing effect of melittin observed by HS-AFM. a** HS-AFM images of extracted *E. coli* lipids immediately after melittin treatment (16 µg/mL). The detection time is indicated on the image. **b** Time evolution of the pore depth and size of the five pores shown in (**a**). **c** HS-AFM images of lipid deposition on the free mica surface after melittin treatment. See Supplementary Movie 9.

using cryo-ET at nanometer resolution. Both pepD2M and melittin share the same ability to produce pores on the membrane which consequently leads to cell shrinkage (Supplementary Fig. 4). The main difference between these two peptides is that pepD2M has a strong lipid-removing ability, hence we could observe low-density membrane, large pores, and lots of lipid clusters. The membrane-removing effect is similar to the membrane-dissolving phenomenon brought about by the detergent. Therefore, we propose that pepD2M functions via a carpet/detergent-like mechanism (Fig. 8). The cell lysis reaction induced by pepD2M is so rapid that the minicell solution became turbid immediately upon the addition of pepD2M. The fast membrane-disruption mechanism diminishes the possibility of the development of drug resistance, making this kind of membrane-disrupting AMPs attractive for combating drug-resistant superbugs. Beyond the contribution of this work to the field of drug development, this research also provides a framework for future investigations of the real-time high-resolution monitoring of membrane disruption by other species (e.g., other AMPs and detergents).

## Methods
### Minicell preparation
*E. coli* (MC4100 *malT*[C] *DmalE malG501 DminCDE::kan*), provided by Prof. Yi-Wei Chang, was cultured in 200 mL of LB medium supplemented with Kanamycin (40 µg/mL) at 37 °C overnight. To collect the minicells, the overnight culture was first centrifuged at 1000×*g* for 10 min at 4 °C before the supernatant was collected and centrifuged again at 3000×*g* for 20 min at 4 °C. Next, the supernatant was collected and concentrated to 5 mL by ultrafiltration at 800×*g* for 50 min at 4 °C using an Amicon® Ultra-15 centrifugal filter (100 kDa cutoff; Merck). The number of minicells in the concentrate was about $6.7 \times 10^7/\mu L$ (counted under a microscope). The cell density was higher than that used in the broth microdilution method for minimum inhibition concentration (MIC) measurement because a considerable number of minicells could be lost during the blotting step of the grid preparation process. Therefore, higher concentrations of AMPs were used in cryo-ET studies.

### Cryo-ET grid preparation
PepD2M (sequence Myr-WKKLKKLLKKLKKL-NH₂; Myr: myristylation; molecular mass: 2003 Da) was synthesized by solid-phase peptide synthesis[12,58]. Melittin (molecular mass: 2846 Da) was purchased from Sigma-Aldrich (Germany). Minicells were mixed with the peptide. A wide range of peptide concentrations (16–640 µg/mL) were tested. An aliquot (~4 µL) of untreated or peptide-treated minicell solution was loaded onto the glow-discharged Quantifoil R 2/2 Holey carbon grid (Quantifoil GmbH, Germany). The grid was then blotted with a filter paper for 3 sec and plunge-frozen in liquid-nitrogen-cooled liquid ethane using a Vitrobot (Mark IV, Thermo Fisher Scientific). The

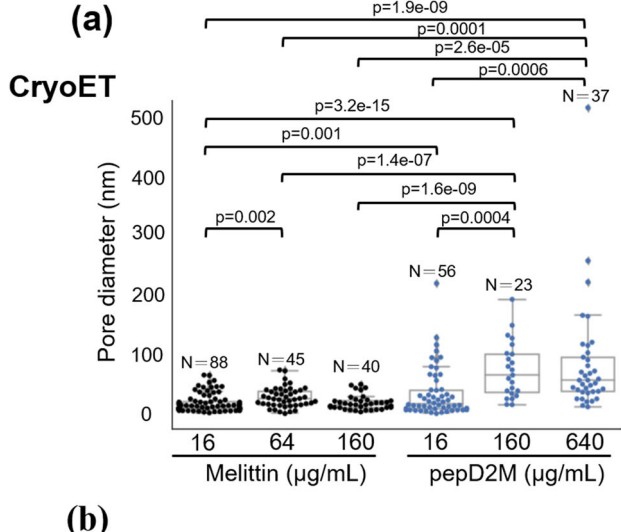

**(a)**

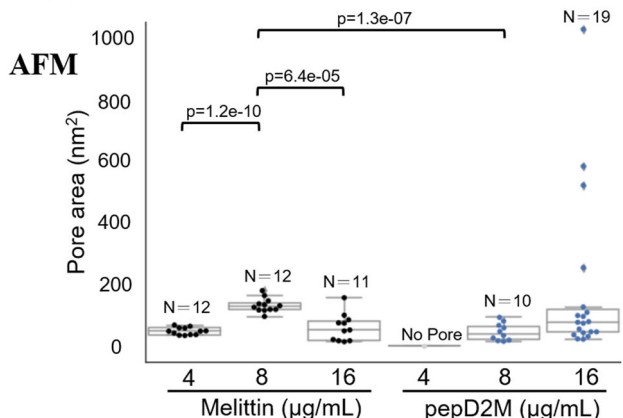

**(b)**

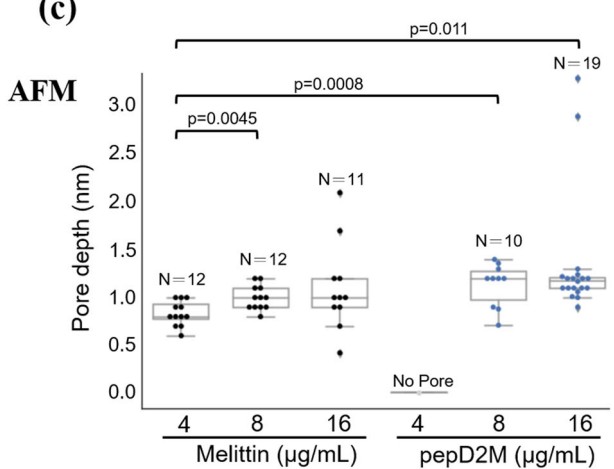

**(c)**

**Fig. 7 | Comparison of the pore area and depth created by pepD2M and melittin. a** Pore diameter measured using cryo-ET. **b** Pore area measured using HS-AFM. **c** Pore depth measured using HS-AFM. Data were analyzed by two-tailed Student's *t*-test. The plots are presented as 25th percentile, median, and 75th percentile, with Tukey whiskers representing ±1.5× the interquartile range. *N* is the number of pores in the images collected from the samples of the indicated concentration. All measurements were taken from distinct samples. For cryo-ET, images were recorded from freshly prepared samples two to three times for each peptide concentration. For HS-AFM, four to five observations were made from freshly prepared samples for each peptide concentration. All measurements were taken from distinct samples.

cryo-EM grid was stored in liquid nitrogen until imaging to prevent devitrification.

**Cryo-electron tomography (cryo-ET)**
Cryo-EM grids were clipped and loaded into a 200 kV Talos Arctica or a 300 kV Titan Krios transmission electron microscope (TEM; Thermo Fisher Scientific). For data collection on Talos Arctica, cryo-EM tilt images were recorded using a Falcon III detector (Thermo Fisher Scientific) operating in linear mode at a magnification of 28,000× with a pixel size of 3.6 Å/pixel. The imaging defocus was set to around −15 μm. Tomographic tilt images were collected using Tomography-4.10.0 software (Thermo Fisher Scientific) with a tilt range of ±60° in 2° increments (from +20° to −60°, and then from +20° to +60°). For each tilt, a dose of -2 e⁻/Å² was used, and the total dose was -120 e⁻/Å² for 61 tilt images. For data collection on Titan Krios, cryo-EM tilt images were recorded with a K3 Summit detector (with GIF Bio-Quantum Energy Filters, Gatan) operating in counting mode at a magnification of 33,000x with a pixel size of 2.75 Å/pixel. The defocus for imaging was set to −6 to −15 μm. Tomographic tilt images were collected using Tomography-5.2.0 software (Thermo Fisher Scientific).

**Tomogram reconstruction and analysis**
The tilt series were binned by a factor of two before alignment and reconstruction into 3D tomograms using the Inspect3D-4.2 software package (Thermo Fisher Scientific). After tilted-image alignment and tilt-axis adjustment, the tomographic reconstruction was performed using the simultaneous iterative reconstruction technique (SIRT) method. To enhance contrast, tomograms were denoised with Topaz v0.2.5[63]. Denoised tomograms were manually segmented using the Amira software package 2022.1 (Thermo Fisher Scientific) with the Membrane Enhancement Filter module. The movies were generated using either the ZAP window of 3dmod within IMOD-4.9.12 or the Amira software package. The distance from OM to PG and the pore diameter were manually measured using the IMOD commands imodinfo in the IMOD slicer windows. To analyze the pixel values across a line profile through OM, PG, and IM in minicell. A 7.2 nm thick tomographic image was saved as TIFF and imported into FIJI/ImageJ 1.52r[64] package. A line (10 pixels in width) was then drawn across the OM, PG, and IM, and the pixel values along this line profile were measured using FIJI's plot profile tool. The pixel values were inverted by subtracting each value from 255 and were exported for analysis and plotting using Numpy[65], Matplotlib[66], and Seaborn libraries in Python 3.5.

**High-speed atomic force microscopy (HS-AFM)**
Polar lipid extract from *E. coli* (Avanti Polar Lipids) was weighed and dissolved in chloroform, and then the solvent was evaporated under N₂ gas. The lipid was then suspended in 5 mM MgCl₂ to a lipid concentration of 1.5 mg/mL and sonicated for a few seconds using a tip-sonicator to form unilamellar lipid vesicles. Next, 2 μL of the liposome solution was deposited onto a freshly cleaved mica surface. After incubation for 5 min, the sample surface was thoroughly rinsed with pure water. The HS-AFM measurements were carried out using a laboratory-built AFM operated in tapping mode[53,55]. A miniaturized cantilever (BL-AC10DS: Olympus, Tokyo, Japan) with a spring constant of -0.1 N m⁻¹, a quality factor of approximately 2, and a resonant frequency of -0.5 MHz in a solution was used. During the HS-AFM imaging of the lipid bilayers, the pepD2M and melittin solutions were injected into the water to a final concentration 4, 8, and 16 μg/mL, and the changes occurring in the membrane were monitored at room temperature. The scanning speed was one frame per second (fps). The images were analyzed with a custom-made tool based on Igor Pro 9. The pixels used in the image recording and the details of imaging processing and analysis are reported in the Supplementary materials.

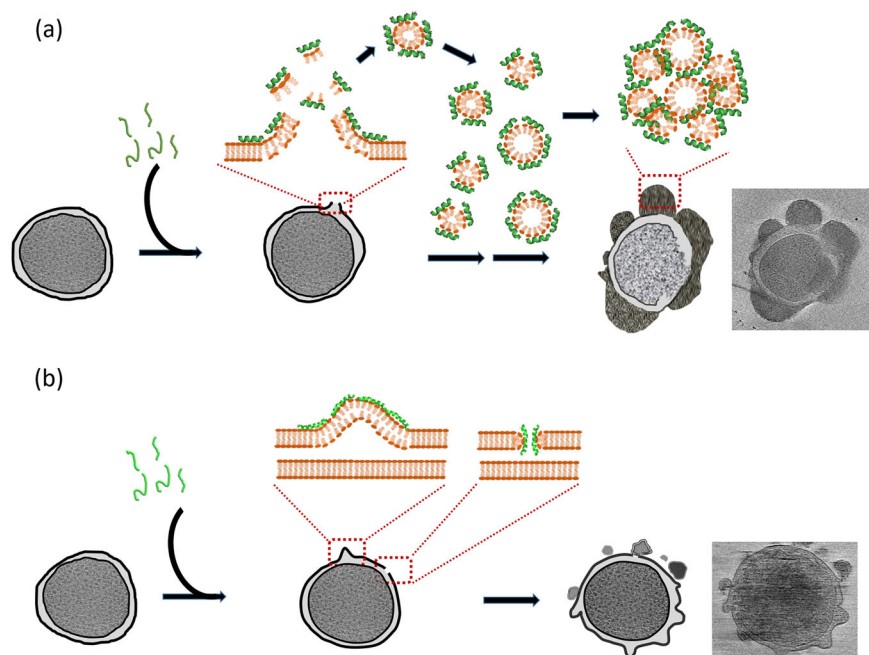

**Fig. 8 | AMP mechanism models. a** The carpet/detergent-like model of the antimicrobial peptide pepD2M. **b** The membrane-thinning model and the pore-forming model of melittin.

## Reporting summary

Further information on research design is available in the Nature Portfolio Reporting Summary linked to this article.

## Data availability

The data that support this study are available from the corresponding authors upon request. The underlying data of Figs. 1, 2, 5, and 7 and Supplementary Figs. 1, 2, and 4 are provided in the Source Data file. The cryo-ET images are deposited in the Figshare repository [https://figshare.com/s/311c7b0c9461ed67cd36] and 3D segmentation models generated from these images are available from the corresponding author. Requests will be answered in 2 weeks. Source data are provided with this paper.

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

## Acknowledgements

This work was funded by Academia Sinica and the Ministry of Science and Technology (MOST) of Taiwan (MOST 108-2321-B-002-052; R. P.-Y. C.). The cryo-EM experiments were performed at the Academia Sinica Cryo-EM Facility (ASCEM). ASCEM is jointly supported by Academia Sinica Core Facility (Innovative Instrument Project; Grant No. AS-CFII-108-110), and the Taiwan Protein Project (Grant No. AS-KPQ-109-TPP2). We thank Dr. Todd Lowary (Institute of Biological Chemistry, Academia Sinica, Taiwan) for his valuable input in manuscript writing and Dr. Po-Yen Lin (Core Facilities, Institute of Cellular and Organismic Biology, Academia Sinica, Taiwan) for his help with using the Amira software. This research was partially supported by Joint Research of the Exploratory Research Center on Life and Living Systems (ExCELLS; ExCELLS program No. 18-101, No. 20-404, No. 22EXC320 and No. 22EXC601). Y.-W.C is supported by the David and Lucile Packard Fellowship for Science and Engineering (2019-69645) and Pennsylvania Department of Health FY19 Health Research Formula Fund.

## Author contributions

E. H.-L.C. conducted all the experiments except HS-AFM and analyzed data; C.-H.W. collected and processed cryo-ET images; Y.-T.L. processed and analyzed cryo-ET images; F.-Y.C., Y.K., T.U., and K.K. performed HS-AFM experiment and analyzed images; L.L. and Y.-W.C. contributed to minicell preparation; M.-C.H. contributed to imaging processing; Y.-W.C. and M.-C.H. commented and revised the manuscript; R.P.-Y.C. designed the project, analyzed data, and wrote and revised the manuscript. All the authors approve the final version.

## Competing interests

The authors declare no competing interests.
