## [Peer Review File · Nature Communications]

Visualizing the membrane disruption action of antimicrobial peptides by cryo-electron tomographyReviewers' Comments:

Reviewer #1:

Remarks to the Author:

This is an original study that uses the technique of cryo-electron tomography (cryo-ET) on E coli minicells for the analysis of the antimicrobial action of an AMP. Moreover, the work uses a complementary technique for the assessment of the activity of the AMP the high-speed atomic force microscopy (hs-afm) on supported lipid bilayers. The study focuses on a newly developed AMP, the pepD2M. In addition, the authors also look at the activity of a well-known AMP, the melittin, with their cryo-ET E coli minicells approach, but in this case, they do not combine explore the action of the melittin with the hs-afm (Fact, that creates an unbalanced design-of-experiment between the pepD2M and the melittin).

Critically, the assay of cryo-ET on E.Coli minicells is not new in the literature. The authors should have mentioned previous works. They should have also mentioned the application of the approach for the assessment of the activity of antimicrobials (checkpoint 2) in the list of remarks below).

The authors do a fair attempt to build a study of correlative microscopy combining cryo-ET and hs-afm. They provide two sets of information. Using both techniques they observe that the pepD2M creates pores in the bacterial membranes. Interestingly, the dimensions of the pores they observe using each technique are completely different; using the cryo-ET the pores are of irregular shape and large dimensions over 100nm, while using the hs-afm the pores are circular and of small >20nm diameter. (The authors do not comment on this, and they should have). Unfortunately, the authors do not succeed in their correlative study: the information from each technique is redundant. It would have been necessary, for a proper correlative study, that each technique provides a new layer of information that only overlaps partially with the information from the rest of the techniques to draw a more complete picture of the action of the sample of the study.

I regret to announce that in the present form I cannot support the publication of this work. The reasons are listed next:

Major issues.

1) The introduction needs severe improvement. It needs to focus more on the subject, to provide a better reference frame. Also, some sentences need rewriting as lines 65-68 and 82-84. Statements in the introduction need to be corrected; for example, line 87 that states that the 'standard biophysical techniques can only reveal macroscopic changes' should say that the standard biophysical techniques rely on extensive averaging, but they can provide molecular resolution information, as it is the case for FRET or NMR. Moreover, in the case of PepD2M which is a de novo-designed drug, it would be useful for the reader if the authors indicated the criteria by which they have selected to study this chemical structure in particular over other possible chemical analogs of the PepD2M. A minor point is that the introduction does not include a summary at the end.

2) The minicells of E.Coli and cryo-ET is not a new approach; hence, the previous literature should be referenced, line 145, for example, the manuscript of J. Liu et al. in 2012 <https://doi.org/10.1073/pnas.1200781109>. Moreover, the approach of cryo-ET on minicells was also used before for the study of the antibiotic action, this should also be mentioned; please check the document '2019_CryoET_Minicells Opto-Antibiotic_proceedings.pdf' that was attached to this review.

3) In the cryo-ET sections, a more complete and statistical description of the shape of the holes that appear in the membrane should be provided.

4) Line 195. The comparison of the data from the authors with the data from Reference 38 by SEM should be clarified. As the depressions shown in the Ref 38 obtained by SEM are not similar to the holes observed by the authors by cryo-ET. Importantly, the sample preparation of SEM is different from that of cryo-ET. Finally, two different AMP molecules are compared. All these aspects should be mentioned and analysed before any conclusion.

5) Line 195. As in 4), the comparison of the holes observed by the authors with the data at Reference 39 by AFM should be clarified. The Ref 39 shows that the AMP caerin forms localized holes in the gram- bacteria *Klebsiella pneumoniae* that grow in time. The Reference 39 employs a different drug and a different bacterial strain than the authors do. The authors should acknowledge these differences and comment on them before any conclusion.

6) Line 195. References are missing. the authors should include relevant manuscripts that deal with the study of the action of antibiotics on cells by AFM as <https://www.nature.com/articles/nnano.2010.29>

7) At the HS-AFM section, the authors must describe the parameters of acquisition of their HS-AFM videos, such as the scanning speed and the number of pixels.

8) The HS-AFM movies are acquired at a high concentration of pepD2M (16 $\mu\text{g}/\text{mL}$) which is higher than the MIC (4-8 $\mu\text{g}/\text{mL}$). The authors should make clear this point in the text as it is critical to understand the action of the AMPs.

9) The authors correctly state that the 'AMP-induced pore formation in the membrane is highly dynamic' following their HS-AFM observations. Nevertheless, the authors do not correctly interpret the dynamics that they observe. They interpret that the pepD2M creates unstable pores that open and close in time on the lipid bilayers extracted from *E. coli* based on the assumption that the pores remain immobile at fixed locations during the period of the acquisition of their hs-afm videos. Such an assumption is not completely correct; the authors did not consider that the pores tend to diffuse laterally in the membrane propelled by thermal motion; the smaller the pore, the more it will diffuse. Below, I show a sequence of frames from one of the hs-afm movies where pore#5 of Fig. 4 is seen. The authors claim that pore#5 forms and disappear at a fixed location, whereas this sequence below shows that it is plausible that the pore diffuses laterally -instead of disappearing-. The authors must consider the possibility that the pores diffuse laterally in their analysis and interpretation.

10) The authors show us a series of time series where they parametrize the evolution of the pores created by the pepD2M in terms of the deepness and radius of the pore. The authors do not describe the procedure to extract these parameters, this procedure must be well described.

11) The authors did not mention that some frames of their hs-afm videos show small protrusions at the membrane that remain immobile during several frames. The protrusions are best appreciated when several frames are averaged, see below. Moreover, at the location where the protrusions are found, holes appear in the latter frames. These protrusions could be related to the initial stages of the action of pep2DM. Previous HS-AFM imaging of the action of the AMPs observed similar structures during the first stages of the action of the Daptomycin <https://www.nature.com/articles/s41467-020-19710-z>

12) The Design of Experiments is unbalanced. It lacks the assessment by high-speed afm of the action melittin.

Minor issues.

13) The CD ellipticity test section, lines 111-126, needs rewriting.

14) Lines 166-174. The description of the functioning of the Cryo-ET should be in the Introduction section.

15) Line 176. The description of status 3 is insufficient, it does not provide useful information to the reader, it reads 'Part of the membrane has a lower electron density'. The implications of the

observation of status 3 should be clarified.

Reviewer #3:

Remarks to the Author:

The manuscript by Chen et al characterizes the membrane disruption of an antimicrobial peptide called pepD2M. The authors use CD analysis to describe the changes in secondary structure of the peptide in the presence of different buffers and liposomes of varying composition. They detect a difference in the peptide spectra if the peptide is incubated with negatively charged lipids in water or if the peptide is incubated with neutral lipids in buffer. They also observe a turbid solution when the peptide is in the presence of both buffer and negatively charged lipids. While I agree, this is suggestive of extensively burst vesicles, it does not rule out that you also get disruption of DOPC liposome bilayers. For this a dye release assay would be more definitive.

The authors go on to use cryo electron tomography to visualize disruption of bacterial mini cell membranes by PepD2M and high-speed atomic force microscopy to visualize the effects of the peptide on supported lipid bilayers. Previous studies have visualized antimicrobial peptides on liposome model systems; therefore, this work is novel in that they are using bacterial minicells which are more representative model system for Gram negative bacterial cell envelopes. Most of the data they present is qualitative. There are no quantitative measurements for their cryotomography observations, and the AFM measurements of penetration depth are only reported for a few individual instances. There are very few tomograms presented of individual minicells and no histograms presented for the AFM which would provide evidence of how representative their observations are across a bigger sampling.

The authors claim insight into the mode of action of pepD2M, but based on the data provided, I don't know how to distinguish differences in the effects of pepD2M vs melittin, both appear to result in lipid blebs (Fig 2g-k and Fig 5a,b), from variability across different minicells treated with either peptide. I am also not able to judge if pepD2M removes lipids similar to the detergent Triton X-100 based on the single image presented in Fig. 5c.

Other comments:

The authors claim nanometer resolution of their tomograms, though do not provide evidence for this calculation.

Overall response to Reviewers

We would like to thank the Reviewers for their insightful comments on our original submission. We have now revised our manuscript based on their comments. The revision took longer because we collected more than 200 cryo-ET images of minicells treated with different concentrations of pepD2M and melittin, enabling us to have quantitative analysis, as the reviewer suggested. Original Figures 2-6 were revised as new Figures 2-5 and 8. We also collected the HS-AFM images of the melittin treatment for comparison with pepD2M to balance our experimental design, as the reviewer suggested. Figure 6 (HS-AFM of melittin treatment) and Figure 7 (statistical comparison of pepD2M and melittin in pore size and depth) are new. An additional fluorescent leakage assay was added as the new Figure S1. One HS-AFM movie (Movie 9: melittin treatment) was added to the supplementary materials.

Moreover, twenty-two references were added, and a professional English service has revised the manuscript. Following are our point-by-point responses to the comments of the Reviewers. The revisions are highlighted in red in the annotated copy of the revised manuscript.

Point-by-point responses to the Reviewers' comments

Reviewer #1

This is an original study that uses the technique of cryo-electron tomography (cryo-ET) on E coli minicells for the analysis of the antimicrobial action of an AMP. Moreover, the work uses a complementary technique for the assessment of the activity of the AMP the high-speed atomic force microscopy (hs-afm) on supported lipid bilayers. The study focuses on a newly developed AMP, the pepD2M. In addition, the authors also look at the activity of a well-known AMP, the melittin, with their cryo-ET E coli minicells approach, but in this case, they do not combine explore the action of the melittin with the hs-afm (Fact, that creates an unbalanced design-of-experiment between the pepD2M and the melittin).

Critically, the assay of cryo-ET on E.Coli minicells is not new in the literature. The authors should have mentioned previous works. They should have also mentioned the application of the approach for the assessment of the activity of antimicrobials (checkpoint 2) in the list of remarks below).

The authors do a fair attempt to build a study of correlative microscopy combining cryo-ET and hs-afm. They provide two sets of information. Using both techniques they observe that the pepD2M creates pores in the bacterial membranes. Interestingly, the dimensions of the pores they observe using each technique are completely different; using the cryo-ET the pores are of irregular shape and large dimensions over 100nm,

while using the hs-afm the pores are circular and of small >20nm diameter. (The authors do not comment on this, and they should have). Unfortunately, the authors do not succeed in their correlative study: the information from each technique is redundant. It would have been necessary, for a proper correlative study, that each technique provides a new layer of information that only overlaps partially with the information from the rest of the techniques to draw a more complete picture of the action of the sample of the study.

Response

We thank the Reviewer for his/her invaluable and insightful comments. We have extensively revised our manuscript and added 22 references. Although minicells of *E.coli* and cryo-ET are not new, combining minicells and cryo-ET to study the mechanism of AMPs is new, which provide a direct 3D visualization of different membrane disruption mechanism. As suggested, we collected more data and did the statistical analysis and comparison for pepD2M and melittin. Our responses are listed below in a point-by-point manner.

Comment 1

The introduction needs severe improvement. It needs to focus more on the subject, to provide a better reference frame. Also, some sentences need rewriting as lines 65-68 and 82-84. Statements in the introduction need to be corrected; for example, line 87 that states that the ‘standard biophysical techniques can only reveal macroscopic changes’ should say that the standard biophysical techniques rely on extensive averaging, but they can provide molecular resolution information, as it is the case for FRET or NMR. Moreover, in the case of PepD2M which is a de novo-designed drug, it would be useful for the reader if the authors indicated the criteria by which they have selected to study this chemical structure in particular over other possible chemical analogs of the PepD2M. A minor point is that the introduction does not include a summary at the end.

Response

We thank the Reviewer’s suggestions. We have extensively revised the introduction and focus on the application of cryo-ET in the direct observation of the membrane disruption of two antimicrobial peptides. We have explained why we chose pepD2M as our target AMP. See pages 4-8.

Comment 2

The minicells of *E. coli* and cryo-ET is not a new approach; hence, the previous

literature should be referenced, line 145, for example, the manuscript of J. Liu et al. in 2012 <https://doi.org/10.1073/pnas.1200781109>. Moreover, the approach of cryo-ET on minicells was also used before for the study of the antibiotic action, this should also be mentioned; please check the document ‘2019_CryoET_Minicells Opto-Antibiotic_proceedings.pdf’ that was attached to this review.

Response

We thank the reviewer’s suggestion and have added more references in the introduction accordingly. Please see pages 5–7. Although minicells of *E.coli* and cryo-ET are not new, combining minicells and cryo-ET to study the mechanism of AMPs is new. We could not find any literature in this regard. We thank the Reviewer for enthusiastically providing us with a pdf file. Unfortunately, we did not see the document ‘2019_CryoET_Minicells Opto-Antibiotic_proceedings.pdf’ attached. We have enquired with the editor about this document but have received no response.

Comment 3

In the cryo-ET sections, a more complete and statistical description of the shape of the holes that appear in the membrane should be provided.

Response

We thank the Reviewer for this excellent suggestion. In this revision, we have accordingly spent a substantial amount of time collecting more images for statistical analysis. We analyzed the morphological changes from the cryo-ET images of the minicells treated with pepD2M and reported the results in the new Fig. 2m and Fig. 2n. We analyzed the morphological changes from the cryo-ET images of the minicells treated with melittin and reported the results in the new Fig. 5g and Fig. 5h. The sizes of the pores induced by pepD2M and melittin were compared in the new Fig 7a. The disruption is statically different, which is consistent with our original statement.

Comment 4

Line 195. The comparison of the data from the authors with the data from Reference 38 by SEM should be clarified. As the depressions shown in the Ref 38 obtained by SEM are not similar to the holes observed by the authors by cryo-ET. Importantly, the sample preparation of SEM is different from that of cryo-ET. Finally, two different AMP molecules are compared. All these aspects should be mentioned and analysed before any conclusion.

Response

PepD2M damages the outer membrane and that sometimes led to a locally concaved cell surface, as shown in Fig. 3f. The Fig. 1e (right) in the original reference 38 also showed a concave cell surface. We have therefore described such similarity in our original manuscript. However, we agree with the Reviewer that comparing images acquired by different imaging technologies is difficult. We have therefore deleted the sentence in the revised manuscript.

Comment 5

Line 195. As in 4), the comparison of the holes observed by the authors with the data at Reference 39 by AFM should be clarified. The Ref 39 shows that the AMP caerin forms localized holes in the gram- bacteria *Klebsiella pneumoniae* that grow in time. The Reference 39 employs a different drug and a different bacterial strain than the authors do. The authors should acknowledge these differences and comment on them before any conclusion.

Response

We agreed with comment 4 from the Reviewer and decided not to compare our results with the data in references 38 and 39 since the peptides, bacteria, and techniques are different.

Comment 6

Line 195. References are missing. the authors should include relevant manuscripts that deal with the study of the action of antibiotics on cells by AFM as <https://www.nature.com/articles/nnano.2010.29>

Response

In fact, this paper (<https://www.nature.com/articles/nnano.2010.29>) was cited as our reference 42 in the original manuscript. This reference is reference 54 in this revised manuscript.

Comment 7

At the HS-AFM section, the authors must describe the parameters of acquisition of their HS-AFM videos, such as the scanning speed and the number of pixels.

Response

We have now added the scanning speed (one frame per second) and the numbers of pixels used in different samples in the supplemental material.

Comment 8

The HS-AFM movies are acquired at a high concentration of pepD2M (16 $\mu\text{g}/\text{mL}$) which is higher than the MIC (4-8 $\mu\text{g}/\text{mL}$). The authors should make clear this point in the text as it is critical to understand the action of the AMPs.

Response

We thank the reviewer's suggestion. In this revision, we collected the HS-AFM images for the 4, 8, and 16 $\mu\text{g}/\text{mL}$ peptide concentrations and reported a statistical comparison in the new Fig. 7b, c. At 4 $\mu\text{g}/\text{mL}$ of pepD2M, we did not observe any membrane disruption. It should be noted that the MIC (minimum inhibition concentration) measurement is the peptide concentration to inhibit bacterial growth. It could not be correlated to the required peptide concentration for pore formation in the AFM study.

Comment 9

The authors correctly state that the 'AMP-induced pore formation in the membrane is highly dynamic' following their HS-AFM observations. Nevertheless, the authors do not correctly interpret the dynamics that they observe. They interpret that the pepD2M creates unstable pores that open and close in time on the lipid bilayers extracted from *E. coli* based on the assumption that the pores remain immobile at fixed locations during the period of the acquisition of their hs-afm videos. Such an assumption is not completely correct; the authors did not consider that the pores tend to diffuse laterally in the membrane propelled by thermal motion; the smaller the pore, the more it will diffuse. Below, I show a sequence of frames from one of the hs-afm movies where pore#5 of Fig. 4 is seen. The authors claim that pore#5 forms and disappear at a fixed location, whereas this sequence below shows that it is plausible that the pore diffuses laterally -instead of disappearing-. The authors must consider the possibility that the pores diffuse laterally in their analysis and interpretation.

Response

Toward addressing this comment, we checked all the AFM images to see whether any pore has lateral movement. We confirmed that no diffusion happens to pore #5. What the Reviewer has seen is the steady device drift during HS-AFM imaging. In Zuttion's paper, they mentioned "it is necessary that the scan speed of the AFM is faster than 1–10 $\mu\text{s}/\text{pixel}$ to attain visualisation of membrane diffusing lipids and small molecules". We scanned at a slower frame rate compared to Zuttion's results. That means that if any diffusion happens during the frames AND returns to its original

position, it would not be observable to us.

Comment 10

The authors show us a series of time series where they parametrize the evolution of the pores created by the pepD2M in terms of the deepness and radius of the pore. The authors do not describe the procedure to extract these parameters, this procedure must be well described.

Response

We thank the Reviewer for the suggestion. We have added the procedures in the supplementary material:

HS-AFM image analysis

The images were analyzed with a custom-made tool based on Igor Pro 9 (<https://www.wavemetrics.com/products/igorpro>). The process involved loading the height data of the AFM image into the software, then subtracting the tilted background using a linear plane function. On the flattened images, a height threshold was defined and used to identify pixels that corresponded to the pores. The area of the pores was calculated by summing up the number of pixels that had a height lower than the threshold, and the maximum depth of the pores was determined from the area of the pores.

Comment 11

The authors did not mention that some frames of their hs-afm videos show small protrusions at the membrane that remain immobile during several frames. The protrusions are best appreciated when several frames are averaged, see below. Moreover, at the location where the protrusions are found, holes appear in the latter frames. These protrusions could be related to the initial stages of the action of pep2DM. Previous HS-AFM imaging of the action of the AMPs observed similar structures during the first stages of the action of the Daptomycin <https://www.nature.com/articles/s41467-020-19710-z>

Response

We thank the Reviewer for pointing this interesting observation out. We have carefully checked all the HS-AFM images again but only found 2 of such protrusions in the sample with pepD2M treatment at 16 ug/mL, as pointed by the pores #6 and #7 in Fig. 4b. We did not find such phenomena in samples treated with lower pepD2M concentrations, i.e., 8 or 4 ug/mL. As such observation is not universal in our images, we decided to be cautious and not over interpreting it. Such protrusion at membrane

was not found in our images of melittin-treated samples, either. Daptomycin is a cyclic lipopeptide, its monomers would oligomerize, and it works against Gram-positive bacteria. Its acting mechanism on lipids could be different from pepD2M and melittin.

Comment 12

The Design of Experiments is unbalanced. It lacks the assessment by high-speed afm of the action melittin.

Response

We agreed with the Reviewer. In this revision we have studied the melittin action on the *E. coli* lipids by HS-AFM and reported the result as the new Fig 6. With these new results, we compared the membrane pore size and depth from melittin vs. pepD2M treatments in the new Fig. 7b, c.

Comment 13

The CD ellipticity test section, lines 111-126, needs rewriting.

Response

We have extensively revised this paragraph to make it clearer. See pages 8-9.

Comment 14

Lines 166-174. The description of the functioning of the Cryo-ET should be in the Introduction section.

Response

We thank the Reviewer's suggestion. We have moved the description to the Introduction section. See pages 5 and 6.

Comment 15

Line 176. The description of status 3 is insufficient, it does not provide useful information to the reader, it reads 'Part of the membrane has a lower electron density'. The implications of the observation of status 3 should be clarified.

Response

We thank the Reviewer's suggestion. We have changed the term "status" to "morphological change". The whole paragraph was revised as follows:

After treatment with pepD2M, several phenomena were observed and classified as five morphological changes: **Change 1**: Irregular holes formed on the OM (indicated

by a yellow arrow in Fig. 2g); **Change 2:** Minicell shrinkage (indicated by a red arrow in Fig. 2h); **Change 3:** Part of the membrane had a lower electron density (indicated by blue arrows in Fig. 2i), which might be because the peptide removed the lipids and decreased the lipid density on the membranes; **Change 4:** The lipids removed from the minicell membrane collapsed into lipid clusters (indicated by black arrowheads in Fig. 2h-l); **Change 5:** The remaining parts of the minicells (indicated by white arrows in Fig. 2j-l) were trapped in the clumps of lipid clusters. We collected 123 tomograms, and the number of pores, shrunken minicells, and residual minicells buried inside the lipid clusters was counted. Our results showed that, at low peptide concentration (16 µg/mL), only pore formation was observed (Fig. 2m), whereas cell shrinkage and lipid cluster formation were observed when higher peptide concentrations were used (Fig. 2m).

Reviewer #3

Comment 1

The manuscript by Chen et al characterizes the membrane disruption of an antimicrobial peptide called pepD2M. The authors use CD analysis to describe the changes in secondary structure of the peptide in the presence of different buffers and liposomes of varying composition. They detect a difference in the peptide spectra if the peptide is incubated with negatively charged lipids in water or if the peptide is incubated with neutral lipids in buffer. They also observe a turbid solution when the peptide is in the presence of both buffer and negatively charged lipids. While I agree, this is suggestive of extensively burst vesicles, it does not rule out that you also get disruption of DOPC liposome bilayers. For this a dye release assay would be more definitive.

Response

We thank the Reviewer for the great suggestion. We have accordingly conducted a fluorescence leakage assay of the DOPE/DOPG and DOPC liposomes treated by pepD2M and melittin. The results are shown in the supplementary material as the new Fig. S1. The results proved that pepD2M favored disruption of the negatively charged liposomes.

Comment 2

The authors go on to use cryo electron tomography to visualize disruption of bacterial mini cell membranes by PepD2M and high-speed atomic force microscopy to visualize the effects of the peptide on supported lipid bilayers. Previous studies have visualized

antimicrobial peptides on liposome model systems; therefore, this work is novel in that they are using bacterial minicells which are more representative model system for Gram negative bacterial cell envelopes. Most of the data they present is qualitative. There are no quantitative measurements for their cryotomography observations, and the AFM measurements of penetration depth are only reported for a few individual instances. There are very few tomograms presented of individual minicells and no histograms presented for the AFM which would provide evidence of how representative their observations are across a bigger sampling.

Response

We thank the Reviewer for his/her suggestion. In this revision, we have conducted more experiments and analyzed the data quantitatively. We collected more cryo-ET and HS-AFM images after treating the cells with different concentrations of antimicrobial peptides. In total, we collected 123 tomograms for pepD2M-treated cells and 141 tomograms for melittin-treated ones. We performed statistical analysis for the count of minicells in different morphological changes. The result from bigger sampling is consistent with our original claim. Please see revised Figures 2 and 5, and the newly added Figure 7. We also conducted HS-AFM for melittin-treated sample and compared the pore size and depth produced by different concentrations of pepD2M and melittin.

Comment 3

The authors claim insight into the mode of action of pepD2M, but based on the data provided, I don't know how to distinguish differences in the effects of pepD2M vs melittin, both appear to result in lipid blebs (Fig 2g-k and Fig 5a,b), from variability across different minicells treated with either peptide. I am also not able to judge if pepD2M removes lipids similar to the detergent Triton X-100 based on the single image presented in Fig. 5c.

Response

PepD2M and melittin produced different consequences. PepD2M produced lots of lipid clusters, and many residuals of minicells were trapped in the clusters, as shown in Fig.2 j-k. For melittin, the surface of the clusters seems to contain double membranes. They look like a cluster of residual minicells (Fig.5d-f). Most importantly, only melittin caused minicells to have blebs on the outer membrane. The bleb is protruded outer membrane, not a lipid cluster adhering to minicells. To help reader to distinguish these differences, we added Fig. 2m, n (for pepD2M) and Fig. 5g, h (for melittin) to summarize those changes observed from more than 250 tomograms. The new figures clearly show the different effects between pepD2M and melittin.

Initially, we wanted to demonstrate pepD2M is a “detergent-like” mechanism by comparing the action of Triton X-100. However, Triton X-100 dissolves lipids too quickly and there were only a limited number of partially-dissolved cells could be preserved for imaging, which prevents us from collecting many tomograms of Triton-treated minicells. We agree with the Reviewer that such limited number of images would prevent proper judgment of whether Triton X-100 acts similarly to pepD2M in disrupting the cell membranes. In this revision, we have therefore removed the Triton X-100 treatment from the main text to avoid confusion.

Comment 4

The authors claim nanometer resolution of their tomograms, though do not provide evidence for this calculation.

Response

We thank the Reviewer for the comment. To support our claim, in Figures 2e, 2f, S2b and S2c, we can clearly measure the distance between the outer membrane and the peptidoglycan cell wall with an average of ~10.7 nm. This shows our tomograms are in the nanometer resolution scale.

Reviewers' Comments:

Reviewer #1:

Remarks to the Author:

The authors have well improved the manuscript and addressed most of the flaws; the design of the experiment is now balanced; the introduction is better framed in the existing literature; and the correlative analysis between the two techniques cryo-ET and HS-AFM is now quantified. In particular, Figure 7 shows the potential of the combined technical approach, and the correlation of the datasets from two levels of nativeness, which reinforces the significance of the results. Nevertheless, I cannot recommend the acceptance for publication of the manuscript in its current state as several critical improvements are remaining:

i. Very Importantly, the post-treatment of the hs-afm videos and the protocol of analysis is missing, and it must be included! As commented, the new Figure 7 shows the potential of the approach, but without the procedure of measurement of the dimensions of the features, it cannot be validated. The readers must have access to the procedure by which the authors have measured the deepness of the 'holes'/dimples that the AMPs produce in the model membranes. For example, it is necessary to know if the authors corrected for possible drift and oscillations in the Z-piezo position by taking any reference as the surface of the lipid bilayer, moreover, the reader ignores if the authors corrected for the XY-drift motions observed in the Movie9 before they measured the variations of the deepness of the 'pores'/ dimples.

ii. There is a terminology issue. In the field of cell membrane studies the term 'pore' is conventionally associated with those features that permeabilize the cell membrane. In the case the deformation induced by the AMP does not traverse across the membrane, to avoid confusion, it would be advisable to use other terms, for example, 'dimple', this term is used in Zuttion et al. Nat. Comm 2020. Hence, the term 'hole' would be reserved for the event that the deformation induced by the AMP connects the two sides of the membrane. The authors use the term 'hole' with the common meaning used in the field of cell membrane studies at line 295.

Another terminology point is the use of the verb 'penetrate'. The term is ambiguous, it does not imply that a hole that permeabilizes the membrane is formed. The authors could use the term 'pierce' instead, for example.

iii. The authors observe two interesting mechanisms of action that unfortunately are not discussed nor analyzed. First, the volcanic-crater-like type of hole is an interesting observation of the activity of the AMPs, the authors should try to frame it in the existing literature and models of action of the AMPs. Second, in line 298 there is another interesting observation, the repair of the damaged membrane by lipid clusters, to my knowledge this is the first time this mechanism is observed, the authors must analyze the mechanism in the context of the existing knowledge and develop the implications of this nice finding.

iv. The following sentence in line 50 — ,compared with the action of a well-known pore-forming peptide, melittin. — needs rephrasing, it is not clear what the authors are comparing and what is the conclusion.

v. In line 53, remove 'important', it is an overstatement.

vi. In line 206, the authors must quantify what they understand by the term 'large pore'.

vii. In line 264, the sentence 'Unlike pepD2M, much fewer lipid clusters were found in the melittin treatment.' needs to be more clear and more quantified.

viii. The authors find that at 16 $\mu\text{g}/\text{mL}$ the melittin does not form pores that pierce the membrane. This finding is interesting, the authors should comment on it in detail as it has implications for the

antimicrobial activity of melittin.

ix. Fig.6a. The spatial dimension scale bar is missing.

x. Fig.7. The lateral dimension of the pores is measured in 'diameter' for the cryo-ET dataset and in 'area' for the HS-AFM dataset. Provided the irregular, non-circular, shapes of the pores that the authors observe, the measurement in terms of 'area' is more appropriate than the 'diameter'. The authors must unify the unit of measurement of the pore lateral dimensions in cryo-ET and HS-AFM.

Reviewer #3:

Remarks to the Author:

The revised manuscript is much improved over the initial submission. I still have some issues/concerns over the statistical rigour in which apply to their data. The only statistical analysis presented is in the pore size, depth.

It is good that they articulate the morphological changes that occur when minicells are treated with the peptide (Changes 1-5). However, these changes aren't then quantified or measured in their larger dataset. Instead, for pepD2M they present quantification of pores, shrunken minicells and residual cell in lipid cluster. If the goal of the analysis is to show that the consequences of treating with pepD2M and melittin on mini cell morphology is different, then the same set of assessment criteria should be applied in both cases. For melittin, they report on pores, minicells with blister, and shrunken minicells. It is also not clear from the legend what is being reported in panels G and H of Fig 5 and M/N of Fig 2.

I am satisfied with the results of the dye release assay to complement their CD analysis.

Overall response to Reviewers

We thank the Reviewers for recognizing our effort on the last revision. Based on the comments, we have revised our manuscript accordingly. We revised Fig. 2 m & 2n, Fig. 5b, 5g&5h, Fig. 6a, Fig. 8a&8b, and Movie 9. We added new Fig. S4 to compare the induced results of pepD2M and melittin at the same concentration in the supplementary materials. Following are our point-by-point responses to their further comments for the revised manuscript. The revisions are highlighted in red in the annotated copy of the second revised manuscript.

Point-by-point responses to the Reviewers' comments

Reviewer #1

Comment i

Very Importantly, the post-treatment of the hs-afm videos and the protocol of analysis is missing, and it must be included! As commented, the new Figure 7 shows the potential of the approach, but without the procedure of measurement of the dimensions of the features, it cannot be validated. The readers must have access to the procedure by which the authors have measured the deepness of the 'holes'/dimples that the AMPs produce in the model membranes. For example, it is necessary to know if the authors corrected for possible drift and oscillations in the Z-piezo position by taking any reference as the surface of the lipid bilayer, moreover, the reader ignores if the authors corrected for the XY-drift motions observed in the Movie9 before they measured the variations of the deepness of the 'pores'/ dimples.

Response

We have added the details of HS-AFM imaging processing and analysis in the revised supplementary material. The drift-corrected Movie 9 is provided in this revision. ***“HS-AFM image processing***

The images were processed and analyzed with a custom-made tool based on Igor Pro 9 (<https://www.wavemetrics.com/products/igorpro>). The acquired AFM raw images were loaded into the software, then subtracting the tilted background using a linear plane function. The image drift in X and Y directions during imaging was corrected by image correlation techniques with static features identified in the image, such as consistent lipid patch boundary or immobilized micelles on the mica or lipid surface. Meanwhile, we did not compensate for drift in the Z direction during the AFM imaging. However, since the dimple depth is measured with respect to the lipid membrane surface, even if there is a drift in the Z direction, we consider that it does not significantly affect the relative height change because the imaging speed is fast enough

compared to the drift to tilt the entire image.

HS-AFM image analysis

As the first step of image analysis, the pixels corresponding to the pores were defined. On the flattened images, the lipid height was determined relative to the mica surface. We then manually defined a height threshold based on the visual appearance of the pores, such that only pixels with heights lower than this threshold were considered to correspond to the pores. The area of the pores was calculated by summing up the number of pixels with a height lower than the threshold. Then, the lowest value determined from the site of the pores was defined as the depth of the pore.”

Comment ii

There is a terminology issue. In the field of cell membrane studies the term 'pore' is conventionally associated with those features that permeabilize the cell membrane. In the case the deformation induced by the AMP does not traverse across the membrane, to avoid confusion, it would be advisable to use other terms, for example, 'dimple', this term is used in Zuttion et al. Nat. Comm 2020. Hence, the term 'hole' would be reserved for the event that the deformation induced by the AMP connects the two sides of the membrane. The authors use the term 'hole' with the common meaning used in the field of cell membrane studies at line 295.

Another terminology point is the use of the verb 'penetrate'. The term is ambiguous, it does not imply that a hole that permeabilizes the membrane is formed. The authors could use the term 'pierce' instead, for example.

Response

We agree that AMP did not transverse across the membrane in most of the sites. Pores #6 and #7 do pierce the membrane. “Dimple” was used in Zuttion et al. Nat. Comm 2020 but not in other AFM papers. To make our description clearer to general readers, we revised the paragraphs as follows:

On page 13

“Fig. 4c shows the time evolution of the pore depth and size for seven pores (#1–#7). Our analysis showed that the AMP-induced pore formation in the membrane is highly dynamic. In the short duration after peptide addition, Pore #4 repeatedly formed and disappeared at the same location (Movie 4). On the other hand, the depth of Pore #2 was almost constant, but the size gradually increased. The maximum depth of Pore #1–#5 is about one nanometer. The small depth indicates that these pores did not pierce the membrane (the thickness of the lipid membrane is about four nanometers). Therefore, they are actually “dimples” rather than “pores”. On the other hand, after

more than five minutes of injecting pepD2M into the solution, the depth and size of the pores formed in the membrane increased (Pores #6 and #7), with the maximum depth being about four nanometers, suggesting that they **pierced** the membrane. Even in such cases, the pore's depth and size fluctuate with time. The lipids removed by the peptide gradually deposited on the free mica surface (Fig. 4d; Movie 6).”

On page 15

“For comparison, we also recorded HS-AFM images of the *E. coli* lipid bilayers treated with melittin. Using the same peptide concentration, melittin formed pores slower (after 553 seconds) than pepD2M (after 16 seconds; Fig. 6a; Movie 9). The pore depth and size of five pores (#8–12) are shown in Fig. 6b. We could also see that a pore appeared and disappeared (Pore #10), and the lipids deposited on the free mica surface (Fig. 6c). The maximum pore depth for most of the pores was around 1 nm, **suggesting that they might be dimples caused by the membrane-thinning effect of melittin rather than toroidal pores which transverse across the membrane.** We did not observe deep lipid penetration (> 3 nm) and large pores (> 200 nm²), as seen in Pore #6 and #7 in the pepD2M treatment (Fig. 4c).”

On page 18

“When comparing the pore size observed through cryo-ET (Fig. 7a), the diameters of the pores created by pepD2M had considerable variation (all the sites with missing membranes were counted as pores) at high concentrations (160 and 640 µg/mL). At the same peptide concentration, the pore sizes created by pepD2M were more extensive than those produced by melittin, suggesting the strong membrane disruption effect of pepD2M. When comparing the pore area observed by HS-AFM (Fig. 7b), melittin did not show concentration-dependent variation. On the other hand, low-concentration pepD2M could not form any pores, while high-concentration pepD2M could create big pores (200–1000 nm²) in the lipid bilayers. We did not compare the pore size observed using cryo-ET and HS-AFM because cryo-ET watched the pore formed on minicells where osmotic pressure plays a role in the membrane damage. According to the pore depth from the HS-AFM data, both peptides created shallow **dimples** on lipids, but high-concentration pepD2M had the chance **to remove the lipid bilayer** (Fig. 7c).”

Comment iii

The authors observe two interesting mechanisms of action that unfortunately are not discussed nor analyzed. First, the volcanic-crater-like type of hole is an interesting observation of the activity of the AMPs, the authors should try to frame it in the existing

literature and models of action of the AMPs. Second, in line 298 there is another interesting observation, the repair of the damaged membrane by lipid clusters, to my knowledge this is the first time this mechanism is observed, the authors must analyze the mechanism in the context of the existing knowledge and develop the implications of this nice finding.

Response

We thank for the reviewer’s suggestion and we totally agreed that volcanic-crater-like type of hole is interesting and lipid cluster, also to our knowledge, is the first time observed. These two novel observations demonstrated the power of cryo-ET in observing AMP effect, which is one of the key points of this work.

The volcanic-crater-like pore is likely not a stable status to be caught easily. It is probably induced by osmotic pressure change. Luckily, we captured the moment in the freezing step. We drew the volcanic-crater-like pore in the revised Figure 8a.

As for the sentence in line 298, we do not mean “the repair of the damaged membrane by lipid.” We revised the sentence as follows on pages 16–17.

“Due to hydrophobic and electrostatic interactions, the lipid clusters can adhere to the cell surface, which might prevent further leakage of cellular content if the adhere site has a pore (Fig. 2h and 2l).”

Fig. 8 (a) The carpet/detergent-like model of the antimicrobial peptide pepD2M. (b)

The membrane-thinning model and the pore-forming model of melittin.

Comment iv

The following sentence in line 50 — ,compared with the action of a well-known pore-forming peptide, melittin. — needs rephrasing, it is not clear what the authors are comparing and what is the conclusion.

Response

We thank the Reviewer's suggestion. We rephrased the sentences as follows on page 3.

“In this study, we use cryo-electron tomography (cryo-ET) and high-speed atomic force microscopy (HS-AFM) to directly visualize the pepD2M-induced disruption of the outer and inner membranes of the Gram-negative bacterium *Escherichia coli*, and compared with a well-known pore-forming peptide, melittin. Our high-resolution cryo-ET images reveal how pepD2M disrupts the *E. coli* membrane using a carpet/detergent-like mechanism.”

Comment v

In line 53, remove 'important', it is an overstatement.

Response

This word was removed in this revision.

Comment vi

In line 206, the authors must quantify what they understand by the term 'large pore'.

Response

We rewrote the sentence as follows on pages 11–12.

“On the membrane of this damaged minicell, three holes could be clearly observed on the OM, while the IM remained intact (inset in Fig. 3a).”

Comment vii

In line 264, the sentence 'Unlike pepD2M, much fewer lipid clusters were found in the melittin treatment.' needs to be more clear and more quantified.

Response

We counted the number of lipid cluster in 140 tomograms. The results are shown in the revised Fig. 5. We revised the sentence as “Unlike pepD2M, there are very few

lipid clusters in the melittin treatment. Only several small lipid clusters (indicated by black arrowheads in Fig. 5b) could be observed. Fig. 5g shows the counts of pores, minicells with blisters, shrunken minicells, and lipid clusters at different peptide concentrations. Their occurrence percentage in 140 tomograms is displayed in Fig. 5h.” on page 15.

Fig. 5 The cryo-tomographic slices and the 3D segmentation of the minicells treated with melittin (a–f). IM: cyan; OM: violet; released material: green. Insets in (a) and (b): enlarged images of the membrane with a pore. The red arrowhead indicates the location of the pore. Lipid clusters are indicated by black arrowheads. The location of blisters and double membranes of shrunken minicells are indicated by white and yellow arrowheads, respectively. Scale bar = 100 nm. (g) Counts of pores, minicells with blisters, shrunken minicells, and lipid clusters after the treatment with different

concentrations of melittin. (h) The proportion of the occurrence of pores, minicells with blisters, shrunken minicells, and lipid clusters in 140 tomograms.

Comment viii

The authors find that at 16 µg/mL the melittin does not form pores that pierce the membrane. This finding is interesting, the authors should comment on it in detail as it has implications for the antimicrobial activity of melittin.

Response

The best way to compare the antimicrobial activity of two AMPs is the bacterial killing assay such as the measurement of Minimal Inhibition Concentration (MIC) or Minimal Bactericidal Concentration (MBC). Whether two peptides at the same concentration can pierce the lipid bilayer on mica cannot be used to compare their antimicrobial activity. The negatively-charged lipid on bacterial surface is a key factor for the bactericidal activity and target selectivity of a positively-charged AMP. However, the lipids on mica is the extracted lipids, not the lipid on the outer membrane.

Comment ix

Fig.6a. The spatial dimension scale bar is missing.

Response

We thank the Reviewer. We added the scale bar in Fig 6a in this revision.

Fig.6 Lipid-removing effect of melittin observed by HS-AFM. (a) HS-AFM images of extracted *E. coli* lipids immediately after melittin treatment (16 $\mu\text{g}/\text{mL}$). The detection time is indicated on the image. (b) Time evolution of the pore depth and size of the five pores shown in (a). (c) HS-AFM images of lipid deposition on the free mica surface after melittin treatment. See Movie 9.

Comment x

Fig.7. The lateral dimension of the pores is measured in ‘diameter’ for the cryo-ET dataset and in ‘area’ for the HS-AFM dataset. Provided the irregular, non-circular, shapes of the pores that the authors observe, the measurement in terms of ‘area’ is more appropriate than the ‘diameter’. The authors must unify the unit of measurement of the

pore lateral dimensions in cryo-ET and HS-AFM.

Response

HS-AFM measures the pixels in the dimples/pores, so we called it “area.” For cryo-ET, we chose the representative slice through the 3D tomogram to capture the opening of the pores, so we called it “diameter.” PepD2M produced lots of holes with various irregular sizes, such as the holes in Fig 3f. Therefore, we used “diameter” to indicate the magnitude of the peptide-induced lesion.

Reviewer #3

Comment

The revised manuscript is much improved over the initial submission. I still have some issues/concerns over the statistical rigour in which apply to their data. The only statistical analysis presented is in the pore size, depth.

It is good that they articulate the morphological changes that occur when minicells are treated with the peptide (Changes 1-5). However, these changes aren't then quantified or measured in their larger dataset. Instead, for pepD2M they present quantification of pores, shrunken minicells and residual cell in lipid cluster. If the goal of the analysis is to show that the consequences of treating with pepD2M and melittin on mini cell morphology is different, then the same set of assessment criteria should be applied in both cases. For melittin, they report on pores, minicells with blister, and shrunken minicells. It is also not clear from the legend what is being reported in panels G and H of Fig 5 and M/N of Fig 2.

I am satisfied with the results of the dye release assay to complement their CD analysis.

Response

We thank the Reviewer's suggestion. A membrane with low density often occurs with other changes, so we did not count it in our last revision. In this revision, we counted changes 1-5. When one minicell has more than one pore, every pore is counted. However, it is difficult to define the range of the membrane with lower density. Therefore, when one minicell having part of the membrane with lower density, we counted “one”. Moreover, the lipid clusters with and without residual cells were counted separately in this revision. See revised Fig2.

According, we revised the paragraph on page 11 as following

“After treatment with pepD2M, several phenomena were observed and classified as five morphological changes: **Change 1:** Irregular holes formed on the OM (indicated by a yellow arrow in Fig. 2g); **Change 2:** Minicell shrinkage (indicated by a red arrow in Fig. 2h); **Change 3:** Part of the membrane had a lower density (indicated by blue arrows in Fig. 2i), which might be because the peptide removed the lipids and decreased the lipid density on the membranes; **Change 4:** The lipids removed from the minicell membrane collapsed into lipid clusters (indicated by black arrowheads in Fig. 2h–l); **Change 5:** The remaining parts of the minicells (indicated by white arrows in Fig. 2j–l) were trapped in the clumps of lipid clusters. We collected 123 tomograms, and these five changes were counted (Fig. 2m). Our results showed that pore formation and membrane with a lower density were observed at a low peptide concentration (16 $\mu\text{g}/\text{mL}$). In contrast, when higher peptide concentrations were used, shrunken cells and lipid clusters with or without residual cells trapped inside were frequently found (Fig. 2n).”

Fig. 2 Cryo-ET studies of minicells with and without pepD2M treatment. (a) A representative square of a Quantifoil R 2/2 Holey carbon grid loaded with minicells. (b) A zoomed-in image of the red box in (a). Minicells could be in a hole or on the carbon film. (c) Two untreated minicells with an enlarged periplasm. (d) A tomographic slice of an untreated minicell. (e) A zoomed-in image of the blue box in (d). OM: outer

membrane; PG: peptidoglycan; IM: inner membrane. (f) The average distance with a value of 10.7 ± 0.1 nm between the OM and PG of eleven minicells was measured using cryo-ET. (g–l) Tomographic slices of minicells treated with pepD2M to show examples of five morphological changes. (g) Change 1: **pore formation** (yellow arrow). (h) Change 2: **shrunken cell** (red arrow). (i) Change 3: **low-density membrane** (blue arrow). (j–l) Change 4: **lipid cluster** (black arrowhead); Change 5: **lipid cluster with cell residuals inside** (white arrow). The scale bar represents 100 nm. The insets in (g) and (h) are the enlarged image from the box. IM: cyan; OM: violet. (m) Counts of **changes 1–5** after the treatment with different concentrations of pepD2M. (n) **The proportion of Changes 1–5 occurrence in 123 tomograms.**

In the melittin treatment, we could only observe pore formation, minicells with blisters, shrunken minicells, and a few lipid clusters. We could not identify a membrane with lower density as we saw in the pepD2M treatment. In this revision, we counted the number of lipid clusters in the revised Fig. 5. We compared the percentage of pore formation, shrunken cells, and lipid clusters in tomograms between pepD2M and melittin in the new Fig. S4. There is no significant difference in pore and shrunken minicell formation. The main difference between these two peptides is in the lipid cluster formation. PepD2M has a strong lipid-removing ability; hence we could observe low-density membrane, large pores, and lots of lipid clusters. That is why we need cryo-ET to distinguish the effect of these two kinds of AMPs.

We revised the paragraph on page 19 as following

“Most importantly, this study is the first to demonstrate the peptide-induced morphological changes of bacterial membranes in a native-like state using cryo-ET at nanometer resolution. **Both pepD2M and melittin share the same ability to produce pores on the membrane which consequently leads to cell shrinkage (Fig. S4). The main difference between these two peptides is that pepD2M has a strong lipid-removing ability, hence we could observe low-density membrane, large pores, and lots of lipid clusters.** The membrane-removing effect is similar to the membrane-dissolving phenomenon brought about by the detergent. Therefore, we propose that pepD2M functions via a carpet/detergent-like mechanism (Fig. 8). The cell lysis reaction induced by pepD2M is so rapid that the minicell solution became turbid immediately upon the addition of pepD2M. The fast membrane-disruption mechanism diminishes the possibility of the development of drug resistance, making this kind of membrane-disrupting AMPs attractive for combating drug-resistant superbugs. Beyond the contribution of this work to the field of drug development, this research also provides a framework for future investigations of the real-time high-resolution monitoring of

membrane disruption by other species (e.g., other AMPs and detergents).”

Fig. 5 The cryo-tomographic slices and the 3D segmentation of the minicells treated with melittin (a–f). IM: cyan; OM: violet; released material: green. Insets in (a) and (b): enlarged images of the membrane with a pore. The red arrowhead indicates the location of the pore. Lipid clusters are indicated by black arrowheads. The location of blisters and shrunken minicells are indicated by white and yellow arrowheads, respectively. Scale bar = 100 nm. (g) Counts of pores, minicells with blisters, shrunken minicells, and lipid clusters after the treatment with different concentrations of melittin. (h) The proportion of the occurrence of pores, minicells with blisters, shrunken minicells, and lipid clusters in 140 tomograms.

Fig. S4 Comparison of pepD2M and melittin. (a) pore formation; (b) shrunken cell formation; (c) lipid cluster.

Reviewers' Comments:

Reviewer #1:

Remarks to the Author:

The study shows the potential of the correlative cryo-ET and hs-afm for pharmaceutical development. The authors have well addressed the flaws that were identified in the previous versions of the manuscript. The reviewer considers this study of interest and is now worthy of publication.

One last request: Because the molecular weight of melittin (2.8kDa) differs from that of pepD2M (1.8kDa), at similar weight-concentration the molar-concentration of pepD2M is 52% higher than that of melittin. The molar concentration is appropriate for the biophysical interpretation even if the weight concentration is common use in medicine and pharmacology, the authors should mention this point in line 323 and could also include it in the captions of the figures that compare the action of melittin and that of pepD2M.

Response to Reviewers

Reviewer #1

Because the molecular weight of melittin (2.8kDa) differs from that of pepD2M (1.8KDa), at similar weight-concentration the molar-concentration of pepD2M is 52% higher than that of melittin. The molar concentration is appropriate for the biophysical interpretation even if the weight concentration is common use in medicine and pharmacology, the authors should mention this point in line 323 and could also include it in the captions of the figures that compare the action of melittin and that of pepD2M.

Response

We thank the reviewer's suggestion. The molecular mass of melittin is 2846 Da. The molecular mass of pepD2M is 2003. At the same weight concentration, the molar concentration of pepD2M is 1.4 fold higher than melittin. It is not a big difference. Anyway, we have emphasized that the concentration is weight concentration on page 18 and added the molecular mass of these two peptides on page 20.